# Imprecise Label Learning: A Unified Framework for Learning with Various Imprecise Label Configurations

## Abstract

Learning with reduced labeling standards, such as noisy label, partial label, and supplementary unlabeled data, which we generically refer to as *imprecise* label, is a commonplace challenge in machine learning tasks. Previous methods tend to propose specific designs for every emerging imprecise label configuration, which is usually unsustainable when multiple configurations of imprecision coexist. In this paper, we introduce imprecise label learning (ILL), a framework for the unification of learning with various imprecise label configurations. ILL leverages expectation-maximization (EM) for modeling the imprecise label information, treating the precise labels as latent variables. Instead of approximating the correct labels for training, it considers the entire distribution of all possible labeling entailed by the imprecise information. We demonstrate that ILL can seamlessly adapt to partial label learning, semi-supervised learning, noisy label learning, and, more importantly, a mixture of these settings, with closed-form learning objectives derived from the unified EM modeling. Notably, ILL surpasses the existing specified techniques for handling imprecise labels, marking the first practical and unified framework with robust and effective performance across various challenging settings. We hope our work will inspire further research on this topic, unleashing the full potential of ILL in wider scenarios where precise labels are expensive and complicated to obtain.

## 1 Introduction

One of the critical challenges in machine learning is the collection of annotated data for model training (He et al., 2016; Vaswani et al., 2017; Devlin et al., 2018; Dosovitskiy et al., 2020; Radford et al., 2021; OpenAI, 2023). Ideally, every data instance would be fully annotated with precise labels. However, collecting such data can be expensive, time-consuming, and error-prone. Often the labels can be intrinsically difficult to ascertain precisely. Factors such as a lack of annotator expertise and privacy concerns can also negatively affect the quality and completeness of the annotations, resulting in incomplete and inaccurate labels.

In an attempt to circumvent this limitation, several methods have been proposed to permit model learning from the data annotated with reduced labeling standards, which are generally easier to obtain. We will refer to such labels as *imprecise.* Fig. 1 illustrates some typical mechanisms of label imprecision that are commonly addressed in the literature. Label imprecision requires a modification of the standard supervised training mechanism to build models for each specific case. For instance, *partial label learning* (PLL) (Cour et al., 2011; Luo & Orabona, 2010; Feng et al., 2020b; Wang et al., 2019a; Wen et al., 2021; Wu et al., 2022; Wang et al., 2022a) allows instances to have a set of candidate labels, instead of a single definitive one. *Semi-supervised Learning* (SSL) (Lee et al., 2013; Samuli & Timo, 2017; Berthelot et al., 2019b;a; Sohn et al., 2020; Xie et al., 2020b; Zhang et al., 2021a; Wang et al., 2022c; 2023; Chen et al., 2023) seeks to enhance the generalization ability when only a small set of labeled data is available, supplemented by a larger unlabeled set. *Noisy label learning* (NLL) (Xiao et al., 2015a; Bekker & Goldberger, 2016; Goldberger & Ben-Reuven, 2016; Ghosh et al., 2017; Han et al., 2018; Zhang & Sabuncu, 2018; Li et al., 2018; Wang et al., 2019c; Liu et al., 2020; Li et al., 2020; Ma et al., 2020b; Zhang et al., 2021c) deals with noisy scenarios where the labels are corrupted or incorrect. There is a greater variety of other forms of label imprecision, including multiple (imprecise) annotations from crowd-sourcing (Ibrahim et al., 2023; Wei et al., 2023a), programmable weak supervision

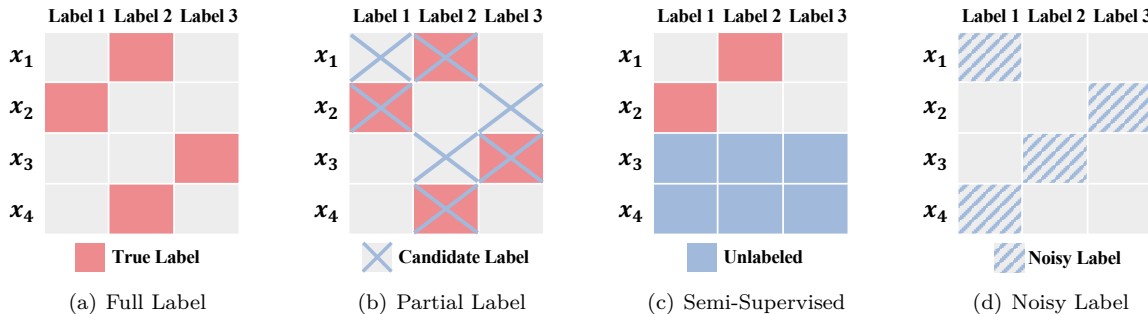

Figure 1: Illustration of the full label and imprecise label configurations. We use an example dataset of 4 training instances and 3 classes. (a) Full label, the annotation is a single true label; (b) Partial label, the annotation is a label candidate set containing true label; (c) Semi-supervised, only part of the dataset is labeled, and the others are unlabeled; (d) Noisy label, the annotation is mislabeled.

(Zhang et al., 2022; Wu et al., 2023a), and bag-level weak supervision (Ilse et al., 2018; Lu et al., 2018; Scott & Zhang, 2020; Zhang et al., 2020; Garg et al., 2021; Feng et al., 2021), among others.

While prior arts have demonstrated success in handling individual configurations of label imprecision, their approaches often differ substantially. They are tailored to a *specific* form of imprecision, as depicted in Fig. 3. Such specificity not only imposes the necessity of devising a solution for emerging types of label imprecision scenarios, but also complicates the deployment in practical settings, where the annotations can be highly complex and may involve *multiple coexisting and interleaved* imprecision configurations. For instance, considering a scenario where both noisy labels and partial labels appear together, it might be challenging to adapt previous methods in NLL or PLL to this scenario since they either rely on the assumption of definite labels (Wei et al., 2023a) or the existence of the correct label among label candidates (Campagner, 2021), thus requiring additional algorithmic design. In fact, quite a few recent works have attempted to address the combinations of imprecise labels in this way, such as partial noisy label (Lian et al., 2022a; Xu et al., 2023) and semi-supervised partial label learning (Wang et al., 2019b; Wang & Zhang, 2020). However, simply introducing a more sophisticated or ad-hoc design can hardly scale to other settings. In addition, most of these approaches attempt to infer the correct labels given the imprecise information (*e.g.* through consistency with adjacent data (Lee et al., 2013; Xie et al., 2020a), iterative refinement (Lv et al., 2020; Arachie & Huang, 2021), average over the given labels (Hüllermeier & Cheng, 2015; Lv et al., 2023), etc., to train the model, which inevitably accumulates error during training, and reduces the generalization performance.

In this paper, we formulate the problem from a different perspective: rather than taking the imprecise label information provided as a potentially noisy or incomplete attempt at assigning labels to instances, we treat it generically as the information that imposes a deterministic or statistical restriction of the actual applicable true labels. We then train the model over the distribution of all possible labeling entailed by the given imprecise information. More specifically, for a dataset with samples $X$ and imprecise label information $I$, we treat the inaccessible full and precise labels $Y$ as a latent variable. The model is then trained to maximize the likelihood of the provided information $I$. Since the likelihood computed over the joint probability $P(X, I; \theta) = \sum_Y P(X, I, Y; \theta)$ must marginalize out $Y$, the actual information $I$ provided could permit a potentially exponential number of labeling. To deal with the resulting challenge of maximizing the logarithm of an expectation, we use the common approach of *expectation-maximization* (EM) (Dempster et al., 1977), where the E-step computes the expectation of $P(X, I, Y; \theta)$ given the posterior of current belief $P(Y|X, I; \theta^t)$ at time step $t$ and the M-step maximizes the tight variational lower bound over $P(X, I; \theta)$. The overall framework is thus largely agnostic to the various nature of label imprecision, with the imprecise label only affecting the manner in which the posterior $P(Y|X, I; \theta^t)$ is computed. In fact, current approaches designed for various imprecise label scenarios can be treated as specific instances of our framework. Thus, our approach can serve as a solution towards a *unified and generalized* view for learning with *various* imprecise labels.

While there exist earlier attempts on generalized or EM solutions for different (other) imprecise supervisions or fuzzy observations (Denœux, 2011; Hüllermeier, 2014; Quost & Denoeux, 2016; Van Rooyen & Williamson, 2017; Gong et al., 2020; Zhang et al., 2020; Chiang & Sugiyama, 2023; Wei et al., 2023b; Xie et al., 2024),

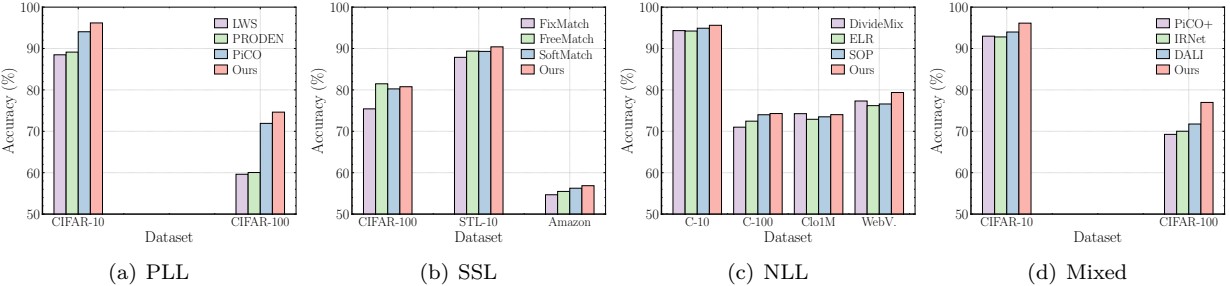

Figure 2: Overview of results comparison to recent SOTA baselines on benchmarks on partial label learning (PLL), semi-supervised learning (SSL), noisy label learning (NLL), and mixed imprecise label learning (MILL). We report the average accuracy over all settings evaluated for each dataset.

they usually require additional assumptions and approximations on the imprecise information for learnablility (Campagner, 2021; 2023), thus presenting limited scalability on practical settings (Quost & Denoeux, 2016). On the contrary, the unified framework we propose subsumes all of these and naturally extends to the more practical "mixed" style of data, where different types of imprecise labels coexist. Moreover, for noisy labels, our framework inherently enables the learning of a *noise model*, as we will show in Section 4.2.

Through comprehensive experiments, we demonstrate that the proposed imprecise label learning (ILL) framework not only outperforms previous methods for dealing with single imprecise labels of PLL, NLL, and SSL, but also presents robustness and effectiveness for mixed imprecise label learning (MILL) settings, leveraging the full potential to more challenging scenarios. Our contributions are summarized as follows:

- We propose an EM framework towards the unification of learning from *various* imprecise labels.
- We establish scalable and consistent state-of-the-art (SOTA) performance with the proposed method on partial label learning, semi-supervised learning, and noisy label learning, demonstrating our method's robustness in more diverse, complex label noise scenarios, as shown in Fig. 2.
- To the best of our knowledge, our work is the first to show the robustness and effectiveness of a single unified method for handling the mixture of various imprecise labels.

## 2 Related Work

Many previous methods have been proposed for dealing with the specific types and some combinations of imprecise label configurations. We revisit the relevant work in this section, especially the state-of-the-art popular baselines for learning with individual and mixture imprecise label configurations.

**Partial label learning (PLL)**. The prior arts can be roughly divided into identification-based for label disambiguation (Zhang & Yu, 2015; Gong et al., 2017; Xu et al., 2019; Wu et al., 2023b) or average-based for utilizing all candidate labels (Hüllermeier & Beringer, 2006; Cour et al., 2011; Lv et al., 2023). The traditional average-based methods usually treat all candidate labels equally, which may involve the misleading false positive labels into training. To overcome these limitations, researchers have explored identification-based methods, viewing the ground-truth label as a latent variable. They seek to maximize its estimated probability using either the maximum margin criterion (Nguyen & Caruana, 2008; Zhang et al., 2016b) or the maximum likelihood criterion (Liu & Dietterich, 2012). Deep learning techniques have recently been incorporated into identification-based methods, yielding promising results across multiple datasets. For example, PRODEN (Lv et al., 2020) proposed a self-training strategy that disambiguates candidate labels using model outputs. CC (Feng et al., 2020b) introduced classifier-consistent and risk-consistent algorithms, assuming uniform candidate label generation. LWS (Wen et al., 2021) relaxed this assumption and proposed a family of loss functions for label disambiguation. More recently, Wang et al. (2022a) incorporated contrastive learning into PLL, enabling the model to learn discriminative representations and show promising results under various levels of ambiguity. RCR involves consistency regularization into PLL recently (Wu et al., 2022).

**Semi-supervised learning (SSL)**. SSL is a paradigm for learning with a limited labeled dataset supplemented by a much larger unlabeled dataset. Consistency regularization and self-training, inspired by clusterness and smoothness assumptions, have been proposed to encourage the network to generate similar predictions for inputs under varying perturbations (Tarvainen & Valpola, 2017; Samuli & Timo, 2017; Miyato et al., 2018). Self-training (Lee et al., 2013; Arazo et al., 2020; Sohn et al., 2020) is a widely-used approach for leveraging unlabeled data. Pseudo Label (Lee et al., 2013; Arazo et al., 2020), a well-known self-training technique, iteratively creates pseudo labels that are then used within the same model. Recent studies focus largely on generating high-quality pseudo-labels. MixMatch (Berthelot et al., 2019b), for instance, generates pseudo labels by averaging predictions from multiple augmentations. Other methods like ReMixMatch (Berthelot et al., 2019a), UDA (Xie et al., 2020a), and FixMatch (Sohn et al., 2020) adopt confidence thresholds to generate pseudo labels for weakly augmented samples, which are then used to annotate strongly augmented samples. Methods such as Dash (Xu et al., 2021b), FlexMatch (Zhang et al., 2021a), and FreeMatch (Wang et al., 2023) dynamically adjust these thresholds following a curriculum learning approach. SoftMatch (Chen et al., 2023) introduces a novel utilization of pseudo-labels through Gaussian re-weighting. SSL has also seen improvements through the incorporation of label propagation, contrastive loss, and meta learning (Iscen et al., 2019; Pham et al., 2021; Li et al., 2021a; Zheng et al., 2022; Wang et al., 2022d).

**Noisy label learning (NLL)**. Overfitting to the noisy labels could result in poor generalization performance, even if the training error is optimized towards zero (Zhang et al., 2016a; 2021b). Several strategies to address the noisy labels have been proposed (Song et al., 2022). Designing loss functions that are robust to noise is a well-explored strategy for tackling the label noise problem (Zhang & Sabuncu, 2018; Wang et al., 2019c; Ma et al., 2020a; Yu et al., 2020). Additionally, methods that re-weight loss (Liu & Tao, 2016) have also been explored for learning with noisy labels. Another common strategy to handle label noise involves assuming that the noisy label originates from a probability distribution that depends on the actual label. Early works (Goldberger & Ben-Reuven, 2016) incorporated these transition probabilities into a noise adaptation layer that is stacked over a classification network and trained in an end-to-end fashion. More recent work, such as Forward (Patrini et al., 2016), prefers to estimate these transition probabilities using separate procedures. However, the success of this method is contingent upon the availability of clean validation data (Northcutt et al., 2021) or additional assumptions about the data (Zhang et al., 2021d). Noise correction has shown promising results in noisy label learning recently (Bai et al., 2021; Li et al., 2021b; 2022; Liu et al., 2022). During the early learning phase, the model can accurately predict a subset of the mislabeled examples (Liu et al., 2020). This observation suggests a potential strategy of correcting the corresponding labels. This could be accomplished by generating new labels equivalent to soft or hard pseudo-labels estimated by the model (Tanaka et al., 2018; Yi & Wu, 2019). Co-Teaching uses multiple differently trained networks for correcting noisy labels (Han et al., 2018). SELFIE (Song et al., 2019) corrects a subset of labels by replacing them based on past model outputs. Another study in (Arazo et al., 2019) uses a two-component mixture model for sample selection, and then corrects labels using a convex combination. Similarly, DivideMix (Li et al., 2020) employs two networks for sample selection using a mixture model and Mixup (Zhang et al., 2017).

**Mixture imprecise label settings**. Various previous works have explored dealing with distinct types of imprecise labels. However, they have yet to tackle a combination of partial labels, limited labels, and noisy labels, which is a highly realistic scenario. For instance, recent attention has been paid to the issue of partial noisy label learning. PiCO+ (Wang et al., 2022b), an extended version of PiCO (Wang et al., 2022a), is tailored specifically for partial noisy labels. IRNet (Lian et al., 2022b) uses two modules: noisy sample detection and label correction, transforming the scenario of noisy PLL into a more traditional PLL. DALI (Xu et al., 2023) is another framework designed to reduce the negative impact of detection errors by creating a balance between the initial candidate set and model outputs, with theoretical assurances of its effectiveness. Additionally, some work has focused on semi-supervised partial label learning (Wang et al., 2019b; Wang & Zhang, 2020). No existing research can effectively address the challenge of handling a combination of partial, limited, and noisy labels simultaneously, which underscores the novelty and significance of our work.

**Previous attempts towards unification of learning from imprecise labels.** There are earlier attempts for the generalized solutions of different kinds of imprecise labels/observations. Denœux (2011) proposed an EM algorithm for the likelihood estimation of fuzzy data and verified the algorithm on linear regression and uni-variate normal mixture estimation. Van Rooyen & Williamson (2017) developed an abstract framework

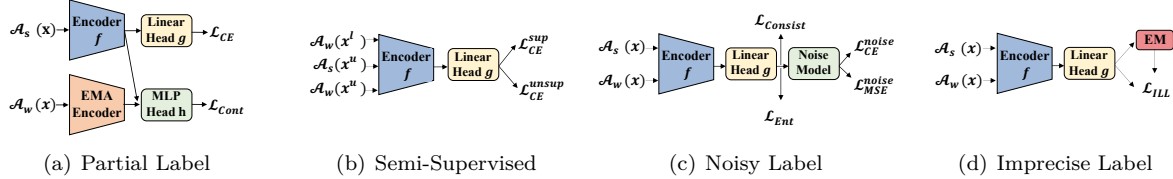

(a) Partial Label        (b) Semi-Supervised        (c) Noisy Label        (d) Imprecise Label

Figure 3: Baseline model pipelines for various imprecise label configurations. (a) PiCO (Wang et al., 2022a) for partial label learning. (b) FixMatch (Sohn et al., 2020) for semi-supervised learning. (c) SOP (Liu et al., 2022) for noisy label learning. (d) The proposed unified framework. It accommodates *any* imprecise label configurations and also mixed imprecise labels with an EM formulation.

that generically tackles label corruption via the Markov transition. Quost & Denoeux (2016) further extended the EM algorithm of fuzzy data on the finite mixture of Gaussians. Gong et al. (2020) proposed a general framework with centroid estimation for imprecise supervision. A unified partial AUC optimization approach was also proposed earlier (Xie et al., 2024). Zhang et al. (2020) and Wei et al. (2023b) proposed generalized solutions for aggregate observations. A unified solution based on dynamic programming for count-based weak supervision was also proposed (Shukla et al., 2023) While relating to these works on the surface, ILL does not require any assumption on the imprecise information and generalizes well to more practical settings with noisy labels. Some other works for individual settings also related EM framework, but usually involved the approximation on the EM (Amini & Gallinari, 2002; Bekker & Goldberger, 2016; Wang et al., 2022a).

## 3 Preliminary

In this section, we illustrate the notations and baselines from different imprecise label settings that adopt various solutions. We will show later how our proposed method generalize and subsumes these prior arts. Let $\mathcal{X}$ denote the input space, and $\mathcal{Y} = [C] := \{1, \ldots, C\}$ represent the label space with $C$ distinct labels. A fully annotated training dataset of size $N$ is represented as $\mathcal{D} = \{(\mathbf{x}_i, y_i)\}_{i \in [N]}$. Learning with imprecise labels involves approximating the mapping function $f \circ g : \mathcal{X} \to \mathcal{Y}$ from a training dataset where the true label $y$ is not fully revealed from the annotation process. Here $f$ is the backbone for feature extraction, $g$ refers to the classifier built on top of the features, and the output from $f \circ g$ is the predicted probability $\mathbf{p}(y|\mathbf{x}; \theta)$, where $\theta$ is the learnable parameter for $f \circ g$. In this study, we primarily consider three imprecise label configurations (as illustrated in Fig. 1) and their corresponding representative learning paradigms (as shown in Fig. 3), namely partial label learning, semi-supervised learning, and noisy label learning.

**Partial label learning (PLL)**. PLL aims to learn with a candidate label set $\mathbf{s} \subset \mathcal{Y}$, where the ground truth label $y \in \mathcal{Y}$ is concealed in $\mathbf{s}$. The training data for partial labels thus becomes $\mathcal{D}_{\mathrm{PLL}} = \{(\mathbf{x}_i, \mathbf{s}_i)\}_{i \in [N]}$. PiCO (Wang et al., 2022a) is a recent contrastive method that employs class prototypes to enhance label disambiguation (as shown in Fig. 3(a)). It optimizes the cross-entropy (CE)[1] loss between the prediction of the augmented training sample $\mathcal{A}_{\mathrm{w}}(\mathbf{x})$ and the disambiguated labels $\hat{\mathbf{s}}$. PiCO learns a set of class prototypes from the features associated with the same pseudo-targets. A contrastive loss, based on MOCO (He et al., 2020), is employed to better learn the feature space, drawing the projected and normalized features $\mathbf{z}_{\mathrm{w}}$ and $\mathbf{z}_{\mathrm{s}}$ of the two augmented versions of data $\mathcal{A}_{\mathrm{w}}(\mathbf{x})$ and $\mathcal{A}_{\mathrm{s}}(\mathbf{x})$ [2] closer. The objective of PiCO is formulated as:

$$\mathcal{L}_{\mathrm{PiCO}} = \mathcal{L}_{\mathrm{CE}}\left(\mathbf{p}(y|\mathcal{A}_{\mathrm{w}}(\mathbf{x}); \theta), \hat{\mathbf{s}}\right) + \mathcal{L}_{\mathrm{Cont}}\left(\mathbf{z}_{\mathrm{w}}, \mathbf{z}_{\mathrm{s}}, \mathcal{M}\right). \tag{1}$$

**Semi-supervised learning (SSL)**. For SSL, we can define the labeled dataset as $\mathcal{D}_{\mathrm{SSL}}^{\mathrm{L}} = \{(\mathbf{x}_i^{\mathrm{l}}, y_i^{\mathrm{l}})\}_{i \in [N^{\mathrm{L}}]}$, and the unlabeled dataset as $\mathcal{D}^{\mathrm{U}} = \{\mathbf{x}_j^{\mathrm{u}}\}_{j \in [N^{\mathrm{L}}+1, N^{\mathrm{L}}+N^{\mathrm{U}}]}$, with $N^{\mathrm{L}} \ll N^{\mathrm{U}}$. A general confidence-thresholding based self-training (Xie et al., 2020a; Sohn et al., 2020) pipeline for SSL is shown in Fig. 3(b). Consider FixMatch (Sohn et al., 2020) as an example; there are usually two loss components: the supervised CE loss on labeled data and the unsupervised CE loss on unlabeled data. For the unsupervised objective, the pseudo-labels $\hat{y}^{\mathrm{u}}$ from the network itself are used to train on the unlabeled data. A "strong-weak" augmentation (Xie et al., 2020a) is commonly adopted. To ensure the quality of the pseudo-labels, only the

---

[1]For simplicity, we use $\mathcal{L}_{\mathbf{CE}}$ for labels of the formats of class indices, one-hot vectors, and class probabilities.
[2]We use $\mathcal{A}_{\mathrm{w}}$ to indicate the weaker data augmentation and $\mathcal{A}_{\mathrm{s}}$ to indicate the stronger data augmentation.

pseudo-labels whose confidence scores $\hat{p}^{\mathrm{u}}$ are greater than a threshold $\tau$ are selected to participate in training:

$$\mathcal{L}_{\mathrm{Fix}} = \mathcal{L}_{\mathrm{CE}} \left( \mathbf{p}(y|\mathcal{A}_{\mathrm{w}}(\mathbf{x}^{\mathrm{l}}); \theta), y^{\mathrm{l}} \right) + \mathbb{1} \left( \hat{p}^{\mathrm{u}} \geq \tau \right) \mathcal{L}_{\mathrm{CE}} \left( \mathbf{p}(y|\mathcal{A}_{\mathrm{s}}(\mathbf{x}^{\mathrm{u}}); \theta), \hat{y}^{\mathrm{u}} \right). \tag{2}$$

**Noisy label learning (NLL)**. NLL aims at learning with a dataset of corrupted labels, $\mathcal{D}_{\mathrm{NLL}} = \{(\mathbf{x}_i, \hat{y}_i)\}_{i \in [N]}$. We illustrate the NLL pipeline (in Fig. 3(c)) with the recent sparse over-parameterization (SOP) model (Liu et al., 2022), where a sparse *noise model* consisting of parameters $\mathbf{u}_i, \mathbf{v}_i \in [-1, 1]^C$ for each sample is adopted. The noise model transforms the network prediction from the true label distribution into the noisy label distribution. A CE loss and a mean-squared-error (MSE) loss optimize parameter $\{\mathbf{u}_i\}$ and $\{\mathbf{v}_i\}$ respectively:

$$\mathcal{L}_{\mathrm{SOP}} = \mathcal{L}_{\mathrm{CE}} \left( \phi \left( \mathbf{p}(y|\mathcal{A}_{\mathrm{w}}(\mathbf{x}); \theta) + \mathbf{m} \right), \hat{y} \right) + \mathcal{L}_{\mathrm{MSE}} \left( \mathbf{p}(y|\mathcal{A}_{\mathrm{w}}(\mathbf{x}); \theta) + \mathbf{m}, \hat{y} \right), \tag{3}$$

where $\phi$ denotes the $L_\infty$ normalization and $\mathbf{m}_i = \mathbf{u}_i \odot \mathbf{u}_i \odot \hat{\mathbf{y}}_i^{\mathrm{oh}} - \mathbf{v}_i \odot \mathbf{v}_i \odot \left( 1 - \hat{\mathbf{y}}_i^{\mathrm{oh}} \right)$, with $\hat{\mathbf{y}}_i^{\mathrm{oh}}$ referring to the one-hot version of $y_i$. Consistency regularization with strong-weak augmentation and entropy class-balance regularization are additionally utilized for better performance in SOP (Liu et al., 2022).

## 4 Imprecise Label Learning

Although current techniques demonstrate potential in addressing particular forms of imprecise labels, they frequently fall short in adaptability and transferability to more complicated and more realistic scenarios where multiple imprecise label types coexist and interleave. This section first defines the proposed expectation-maximization (EM) formulation for learning with various imprecise labels. Then, we demonstrate that our unified framework seamlessly extends to partial label learning, semi-supervised label learning, noisy label learning, and the more challenging setting of mixed imprecise label learning. Connections and generalization to previous pipelines can also be drawn clearly under the proposed EM framework.

### 4.1 A Unified Framework for Learning with Imprecise Labels

**Exploiting information from imprecise labels**. The challenge of learning with imprecise labels lies in learning effectively with inaccurate or incomplete annotation information. Per the analysis above, prior works catering to specific individual imprecise labels either explicitly or implicitly attempt to infer the precise labels from the imprecise label information. For example, partial label learning concentrates on the disambiguation of the ground truth label from the label candidates (Wang et al., 2022a; Lian et al., 2022b; Xu et al., 2023) or averaging equally over the label candidates (Hüllermeier & Beringer, 2006). In semi-supervised learning, after the model initially learns from the labeled data, the pseudo-labels are treated as correct labels and utilized to conduct self-training on the unlabeled data (Arazo et al., 2020; Sohn et al., 2020). Similarly, for noisy label learning, an integral part that helps mitigate overfitting to random noise is the implementation of an accurate noise model capable of identifying and rectifying the incorrect labels (Li et al., 2020; Liu et al., 2022), thereby ensuring the reliability of the learning process. However, inferring the correct labels from the imprecise labels or utilizing the imprecise labels directly can be very challenging and usually leads to errors accumulated during training (Arazo et al., 2020; Chen et al., 2022), which is also known as the confirmation bias. In this work, we take a different approach: we consider all possible labeling along with their likelihood that the imprecise labels fulfill to train the model, rather than using a single rectified label from the imprecise information. Such an approach also eliminates the requirements for designing different methods for various imprecise labels and provides a unified formulation instead, where closed-form solutions can be derived.

**A unified framework for learning with imprecise labels (ILL)**. Let $\{\mathbf{x}_i\}_{i \in [N]}$ represent the features as realizations from $X$ and $\{y_i\}_{i \in [N]}$ represent their precise labels as realizations from $Y$ for the training data. Ideally, $Y$ would be fully specified for $X$. In the imprecise label scenario, however, $Y$ is not provided; instead we obtain imprecise label information $I$. We view $I$ not as *labels*, but more abstractly as a variable representing the *information* about the labels. From this perspective, the actual labels $Y$ would have a distribution $P(Y|I)$, and $I$ can present in various forms. When the information $I$ provided is the precise true label of the data, $P(Y|I)$ would be a delta distribution, taking a value 1 at the true label, and 0 elsewhere. If $I$ represents partial labels, then $P(Y|I)$ would have non-zero value over the candidate labels, and be 0

elsewhere. When $I$ represents a set of noisy labels, $P(Y|I)$ would represent the distribution of the true labels, given the noisy labels. When $I$ does not contain any information, i.e., unlabeled data, $Y$ can take any value.

By the maximum likelihood estimation (MLE) principle, we must estimate the model to maximize the likelihood of the data/information we have been provided, namely $X$ and $I$. Let $P(X, I; \theta)$ represent a parametric form for the joint distribution of $X$ and $I$[3] Explicitly considering the labels $Y$, we have $P(X, I; \theta) = \sum_Y P(X, Y, I; \theta)$. The maximum likelihood principle requires us to find:

$$\theta^* = \arg\max_{\theta} \ \log P(X, I; \theta) = \arg\max_{\theta} \ \log \sum_Y P(X, Y, I; \theta), \tag{4}$$

with $\theta^*$ denotes the optimal value of $\theta$. Eq. (4) features the log of an expectation and cannot generally be solved in closed-form, and requires iterative hill-climbing solutions. Of these, arguably the most popular is the expectation-maximization (EM) algorithm (Dempster et al., 1977), which iteratively maximizes a tight variational lower bound on the log-likelihood. In our case, applying it becomes:

$$\begin{aligned}
\theta^{t+1} &= \arg\max_{\theta} \mathbb{E}_{Y|X,I;\theta^t} \left[\log P(X, Y, I; \theta)\right] \\
&= \arg\max_{\theta} \mathbb{E}_{Y|X,I;\theta^t} \left[\log P(Y|X; \theta) + \log P(I|X, Y; \theta)\right],
\end{aligned} \tag{5}$$

where $\theta^t$ is the $t^{\text{th}}$ estimate of the optimal $\theta$. Note that $P(X; \theta)$ is omitted from Eq. (5) since $P(X)$ does not rely on $\theta$. The detailed derivation of the variational lower bound is shown in Appendix B.1. There are several implications from Eq. (5). (i) The expectation over the posterior $P(Y|X, I; \theta^t)$ equates to considering *all* labeling entailed by the imprecise label information $I$, rather than any single (possibly corrected) choice of label. For independent instances setting studied in this paper, we can derive closed-form training objectives from this formulation as shown in Section 4.2. (ii) The property of the second term $\log P(I|X, Y; \theta)$ is dependent on the nature of imprecise label $I$. If $I$ contains information about the true labels $Y$, such as the actual labels or the label candidates, it can be reduced to $P(I|Y)$, *i.e.*, the probability of $I$ is no longer dependent on $X$ or $\theta$ and thus can be ignored from Eq. (5). If $I$ represents the noisy labels, $P(I|X, Y; \theta)$ instead includes a potentially learnable noise model. (iii) It is a general framework towards the unification of any label configuration, including full labels, partial labels, low-resource labels, noisy labels, etc. In this work, we specialize the proposed EM framework to PLL, SSL, NLL, and the mixture of them in the following.

## 4.2 Instantiating the Unified EM Formulation

We illustrate how to seamlessly expand the formulation from Eq. (5) to partial label learning, semi-supervised learning, noisy label learning, and mixture settings, with derived closed-form loss function[4] for each setting here. The actual imprecise labels only affect the manner in which the posterior $P(Y|X, I; \theta^t)$ is computed for each setting. We show that all learning objectives derived from Eq. (5) naturally include a consistency term with the posterior as the soft target. We also demonstrate that the proposed unified EM framework closely connects with the prior arts, which reveals the potential reason behind the success of these techniques. Note that while we only demonstrate the application of the proposed framework to four settings here, it can also be flexibly extended to other settings. More details of derivation below are shown in Appendix B.

**Partial label learning (PLL)**. The imprecise label $I$ for partial labels is defined as the label candidate sets $S$ containing the true labels. These partial labels indicate that the posterior $P(Y|X, S; \theta^t)$ can only assign its masses on the candidate labels. Since $S$ contains the information about the true labels $Y$, $P(S|X, Y; \theta)$ reduces to $P(S|Y)$, and thus can be ignored. We also demonstrate with instance dependent partial labels that maintains $P(S|X, Y; \theta)$ in Appendix C.2.2. Defining the label candidates as $\{\mathbf{s}_i\}_{i \in [N]}$ and substituting it in Eq. (5), we have the loss function of PLL derived using ILL framework:

$$\mathcal{L}_{\text{ILL}}^{\text{PLL}} = -\sum_{Y \in [C]} P(Y|X, S; \theta^t) \log P(Y|X; \theta) \equiv \mathcal{L}_{\text{CE}} \left(\mathbf{p}(y|\mathcal{A}_{\text{s}}(\mathbf{x}); \theta), \mathbf{p}(y|\mathcal{A}_{\text{w}}(\mathbf{x}), \mathbf{s}; \theta^t)\right), \tag{6}$$

---

[3]The actual parameters $\theta$ may apply only to some component such as $P(Y|X; \theta)$ of the overall distribution; we will nonetheless tag the entire distribution $P(X, I; \theta)$ with $\theta$ to indicate that it is dependent on $\theta$ overall.

[4]To formulate the loss function, we convert the problem to minimization of the negative log-likelihood.

where $\mathbf{p}(y|\mathcal{A}_{\mathrm{w}}(\mathbf{x}), \mathbf{s}; \theta^t)$ is the normalized probability that $\sum_{k \in C} p_k = 1$, and $p_k = 0, \forall k \in \mathbf{s}$. Eq. (6) corresponds exactly to consistency regularization (Xie et al., 2020a), with the normalized predicted probability as the soft pseudo-targets. This realization on PLL shares similar insights as (Wu et al., 2022) which exploits a gradually induced loss weight for PLL on multiple augmentations of the data. However, our framework is much simpler and more concise as shown in Appendix C.2.2, which does not require additional techniques.

**Semi-supervised learning (SSL)** In SSL, the input $X$ consists of the labeled data $X^{\mathrm{L}}$ and the unlabeled data $X^{\mathrm{U}}$. The imprecise label for SSL is realized as the limited number of full labels $Y^{\mathrm{L}}$ for $X^{\mathrm{L}}$. The labels $Y^{\mathrm{U}}$ for unlabeled $X^{\mathrm{U}}$ are unknown and become the latent variable. Interestingly, for the unlabeled data, there is no constraint on possible labels it can take. The posterior $P(Y^{\mathrm{U}}|X^{\mathrm{L}}, X^{\mathrm{U}}, Y^{\mathrm{L}}; \theta)$, which is the actual prediction from the network, can be directly utilized as soft targets for self-training. Since $Y^{\mathrm{L}}$ is conditionally independent with $Y^{\mathrm{U}}$ given $X$, the second term of Eq. (5): $P(Y^{\mathrm{L}}|X^{\mathrm{L}}, X^{\mathrm{U}}, Y^{\mathrm{U}}; \theta)$, is reduced to $P(Y^{\mathrm{L}}|X^{\mathrm{L}}; \theta)$, which corresponds to the supervised objective on labeled data. The loss function for SSL thus becomes:

$$
\begin{aligned}
\mathcal{L}_{\mathrm{ILL}}^{\mathrm{SSL}} = &- \sum_{Y \in [C]} P(Y^{\mathrm{U}}|X^{\mathrm{U}}, X^{\mathrm{L}}, Y^{\mathrm{L}}; \theta^t) \log P(Y^{\mathrm{U}}|X^{\mathrm{U}}, X^{\mathrm{L}}; \theta) - \log P(Y^{\mathrm{L}}|X^{\mathrm{L}}; \theta) \\
\equiv & \mathcal{L}_{\mathrm{CE}} \left( \mathbf{p}(y|\mathcal{A}_{\mathrm{s}}(\mathbf{x}^{\mathrm{u}}); \theta), \mathbf{p}(y|\mathcal{A}_{\mathrm{w}}(\mathbf{x}^{\mathrm{u}}); \theta^t) \right) + \mathcal{L}_{\mathrm{CE}} \left( \mathbf{p}(y|\mathcal{A}_{\mathrm{w}}(\mathbf{x}^{\mathrm{l}}); \theta), y^{\mathrm{l}} \right)
\end{aligned}
\tag{7}
$$

The first term corresponds to the unsupervised consistency regularization usually employed in SSL, and the second term refers to the supervised CE loss only on labeled data. Eq. (7) has several advantages over the previous methods. It adopts the prediction as soft-targets of all possible labeling on unlabeled data, potentially circumventing the confirmation bias caused by pseudo-labeling and naturally utilizing all unlabeled data which resolves the quantity-quality trade-off commonly existing in SSL (Sohn et al., 2020; Chen et al., 2023). It also indicates that previous pseudo-labeling with confidence threshold implicitly conducts the EM optimization, where the maximal probable prediction approximates the expectation, and the degree of the approximation is determined by the threshold $\tau$, rationalizing the effectiveness of dynamic thresholding.

**Noisy label learning (NLL).** Things become more complicated here since the noisy labels $\hat{Y}$ do not directly reveal the true information about $Y$, thus $P(\hat{Y}|Y, X; \theta)$ inherently involves a noise model that needs to be learned. We define a simplified instance-independent[5] noise transition model $\mathcal{T}(\hat{Y}|Y; \omega)$ with parameters $\omega$, and take a slightly different way to formulate the loss function for NLL from the ILL framework:

$$
\begin{aligned}
\mathcal{L}_{\mathrm{ILL}}^{\mathrm{NLL}} = &- \sum_{Y \in [C]} P(Y|X, \hat{Y}; \theta^t, \omega^t) \log P(Y|X, \hat{Y}; \theta, \omega^t) - \log P(\hat{Y}|X; \theta, \omega) \\
\equiv & \mathcal{L}_{\mathrm{CE}} \left( \mathbf{p}(y|\mathcal{A}_{\mathrm{s}}(\mathbf{x}), \hat{y}; \theta, \omega^t), \mathbf{p}(y|\mathcal{A}_{\mathrm{w}}(\mathbf{x}), \hat{y}; \theta^t, \omega^t) \right) + \mathcal{L}_{\mathrm{CE}} \left( \mathbf{p}(\hat{y}|\mathcal{A}_{\mathrm{w}}(\mathbf{x}); \theta, \omega), \hat{y} \right),
\end{aligned}
\tag{8}
$$

where the parameters $\omega$ and $\theta$ are learned end-to-end. The first term corresponds to the consistency regularization of prediction conditioned on noisy labels and the second term corresponds to the supervised loss on noisy predictions that are converted from the ground truth predictions. Both quantities are computed using the noise transition model given the noisy label $\hat{y}$:

$$
\mathbf{p}(y|\mathbf{x}, \hat{y}; \theta, \omega^t) \propto \mathbf{p}(y|\mathbf{x}; \theta) \mathcal{T}(\hat{y}|y; \omega^t), \text{and } \mathbf{p}(\hat{y}|\mathbf{x}; \theta, \omega) = \sum_{y \in [C]} \mathbf{p}(y|\mathbf{x}; \theta) \mathcal{T}(\hat{y}|y; \omega).
\tag{9}
$$

**Mixture imprecise label learning (MILL).** We additionally consider a more practical setting, mixture of imprecise label learning, with partial labels, noisy labels, and unlabeled data interleaved together. On the unlabeled data, the unsupervised objective is the same as the unsupervised consistency regularization of SSL as shown in Eq. (7). The labeled data here present partial and noisy labels $\hat{\mathbf{s}}$. Thus the noisy supervised objective in Eq. (9) becomes the supervised consistency regularization as in Eq. (6) of partial label setting to train the noise transition model, and the noisy unsupervised objective becomes the consistency regularization of the prediction conditioned on noisy partial labels. Thus we have the loss function for MILL derived as:

$$
\begin{aligned}
\mathcal{L}_{\mathrm{ILL}}^{\mathrm{MILL}} = & \mathcal{L}_{\mathrm{CE}} \left( \mathbf{p} \left( y \mid \mathcal{A}_{\mathrm{s}}(\mathbf{x}^l), \hat{\mathbf{s}}^l; \theta, \omega^t \right), \mathbf{p} \left( y \mid \mathcal{A}_{\mathrm{w}}(\mathbf{x}^l), \hat{\mathbf{s}}^l; \theta^t, \omega^t \right) \right) \\
& + \mathcal{L}_{\mathrm{CE}} \left( \mathbf{p} \left( \hat{y} \mid \mathcal{A}_{\mathrm{w}}(\mathbf{x}^l); \theta, \omega \right), \hat{\mathbf{s}}^l \right) \\
& + \mathcal{L}_{\mathrm{CE}} \left( \mathbf{p}(y|\mathcal{A}_{\mathrm{s}}(\mathbf{x}^{\mathrm{u}}); \theta), \mathbf{p}(y|\mathcal{A}_{\mathrm{w}}(\mathbf{x}^{\mathrm{u}}); \theta^t) \right)
\end{aligned}
\tag{10}
$$

---

[5]A more complicated instance-dependent noise model $\mathcal{T}(\hat{Y}|Y, X; \omega)$ can also be formulated under our unified framework, but not considered in this work. Also, since we use $\mathcal{T}$ both in forward fashion and backward fashion, it is unidentifiable in this work.

Table 1: Accuracy of different partial ratio $q$ on CIFAR-10, CIFAR-100, and CUB-200 for **partial label learning**. The best and the second best results are indicated in **bold** and underline respectively.

| Dataset | CIFAR-10 | | | CIFAR-100 | | | CUB-200 |
|---|---|---|---|---|---|---|---|
| Partial Ratio $q$ | 0.1 | 0.3 | 0.5 | 0.01 | 0.05 | 0.1 | 0.05 |
| Fully-Supervised | | $94.91_{\pm0.07}$ | | | $73.56_{\pm0.10}$ | | - |
| LWS (Wen et al., 2021) | $90.30_{\pm0.60}$ | $88.99_{\pm1.43}$ | $86.16_{\pm0.85}$ | $65.78_{\pm0.02}$ | $59.56_{\pm0.33}$ | $53.53_{\pm0.08}$ | $39.74_{\pm0.47}$ |
| PRODEN (Lv et al., 2020) | $90.24_{\pm0.32}$ | $89.38_{\pm0.31}$ | $87.78_{\pm0.07}$ | $62.60_{\pm0.02}$ | $60.73_{\pm0.03}$ | $56.80_{\pm0.29}$ | $62.56_{\pm0.10}$ |
| CC (Feng et al., 2020b) | $82.30_{\pm0.21}$ | $79.08_{\pm0.07}$ | $74.05_{\pm0.35}$ | $49.76_{\pm0.45}$ | $47.62_{\pm0.08}$ | $35.72_{\pm0.47}$ | $55.61_{\pm0.02}$ |
| MSE (Feng et al., 2020a) | $79.97_{\pm0.45}$ | $75.65_{\pm0.28}$ | $67.09_{\pm0.66}$ | $49.17_{\pm0.05}$ | $46.02_{\pm1.82}$ | $43.81_{\pm0.49}$ | $22.07_{\pm2.36}$ |
| EXP (Feng et al., 2020a) | $79.23_{\pm0.10}$ | $75.79_{\pm0.21}$ | $70.34_{\pm1.32}$ | $44.45_{\pm1.50}$ | $41.05_{\pm1.40}$ | $29.27_{\pm2.81}$ | $9.44_{\pm2.32}$ |
| PiCO (Wang et al., 2022a) | $\underline{94.39_{\pm0.18}}$ | $\underline{94.18_{\pm0.12}}$ | $\underline{93.58_{\pm0.06}}$ | $\underline{73.09_{\pm0.34}}$ | $\underline{72.74_{\pm0.30}}$ | $\underline{69.91_{\pm0.24}}$ | $\mathbf{72.17_{\pm0.72}}$ |
| Ours | $\mathbf{96.37_{\pm0.08}}$ | $\mathbf{96.26_{\pm0.03}}$ | $\mathbf{95.91_{\pm0.05}}$ | $\mathbf{75.31_{\pm0.19}}$ | $\mathbf{74.58_{\pm0.03}}$ | $\mathbf{74.00_{\pm0.02}}$ | $\underline{70.77_{\pm0.29}}$ |

We can compute both quantity through the noise transition model:

$$\mathbf{p}(y|\mathbf{x},\hat{\mathbf{s}};\theta,\omega^t) \propto \mathbf{p}(y|\mathbf{x};\theta)\prod_{\hat{y}\in\hat{\mathbf{s}}}\mathcal{T}(y|\hat{y};\omega^t), \text{and } \mathbf{p}(\hat{y}|\mathbf{x};\theta,\omega) = \sum_{y\in[C]}\mathbf{p}(y|\mathbf{x};\theta)\mathcal{T}(\hat{y}|y;\omega). \qquad (11)$$

## 5 Experiments

In this section, we conduct extensive experiments to evaluate ILL. Albeit simple, the ILL framework achieves comparable state-of-the-art performance regarding previous methods on partial label learning, semi-supervised learning, and noisy label learning. Moreover, our experiments show that ILL could be easily extended to a more practical setting with a mixture of various imprecise label configurations. For all settings, we additionally adopt an entropy loss for balancing learned cluster sizes (John Bridle, 1991; Joulin & Bach, 2012), similarly as (Liu et al., 2022; Wang et al., 2023). Experiments are conducted with three runs using NVIDIA V100 GPUs.

### 5.1 Partial Label Learning

**Setup**. Following (Wang et al., 2022a), we evaluate our method on partial label learning setting using CIFAR-10 (Krizhevsky et al., 2009), CIFAR-100 (Krizhevsky et al., 2009), and CUB-200 (Welinder et al., 2010). We generate partially labeled datasets by flipping negative labels to false positive labels with a probability $q$, denoted as a partial ratio. The $C-1$ negative labels are then uniformly aggregated into the ground truth label to form a set of label candidates. We consider $q \in \{0.1, 0.3, 0.5\}$ for CIFAR-10, $q \in \{0.01, 0.05, 0.1\}$ for CIFAR-100, and $q = 0.05$ for CUB-200. We choose six baselines for PLL using ResNet-18 (He et al., 2016): LWS (Wen et al., 2021), PRODEN (Lv et al., 2020), CC (Feng et al., 2020b), MSE and EXP (Feng et al., 2020a), and PiCO (Wang et al., 2022a). The detailed hyper-parameters, comparison with the more recent method R-CR (Wu et al., 2022) that utilizes a different training recipe and model (Zagoruyko & Komodakis, 2016), and comparison with instance-dependent partial labels (Xu et al., 2021a) are shown in Appendix C.2.2.

**Results**. The results for PLL are shown in Table 1. Our method achieves the best performance compared to the baseline methods. Perhaps more surprisingly, on CIFAR-10 and CIFAR-100, our method even outperforms the fully-supervised reference, indicating the potential better generalization capability using the proposed framework, sharing similar insights as in Wu et al. (2022). While PiCO adopts a contrastive learning objective, our method still surpasses PiCO by an average of **2.13**% on CIFAR-10 and **2.72**% on CIFAR-100. Our approach can be further enhanced by incorporating contrastive learning objectives, potentially leading to more significant performance improvements as in the ablation study of PiCO (Wang et al., 2022a).

### 5.2 Semi-Supervised Learning

**Setup**. For experiments of SSL, we follow the training and evaluation protocols of USB (Wang et al., 2022d) on image and text classification. To construct the labeled dataset for semi-supervised learning, we uniformly select $l/C$ samples from each class and treat the remaining samples as the unlabeled dataset. We present the results on CIFAR-100 and STL-10 (Krizhevsky et al., 2009) for image classification, and IMDB (Maas et al., 2011) and Amazon Review (McAuley & Leskovec, 2013) for text classification. We compare with the current

Table 2: Error rate of different number of labels $l$ on CIFAR-100, STL-10, IMDB, and Amazon Review datasets for **semi-supervised learning**.

| Datasets | CIFAR-100 | | STL-10 | | IMDB | | Amazon Review | |
|---|---|---|---|---|---|---|---|---|
| # Labels $l$ | 200 | 400 | 40 | 100 | 20 | 100 | 250 | 1000 |
| AdaMatch (Berthelot et al., 2021) | $22.32_{\pm1.73}$ | $16.66_{\pm0.62}$ | $13.64_{\pm2.49}$ | $\underline{7.62}_{\pm1.90}$ | $8.09_{\pm0.99}$ | $\underline{7.11}_{\pm0.20}$ | $45.40_{\pm0.96}$ | $\mathbf{40.16_{\pm0.49}}$ |
| FixMatch (Sohn et al., 2020) | $29.60_{\pm0.90}$ | $19.56_{\pm0.52}$ | $16.15_{\pm1.89}$ | $8.11_{\pm0.68}$ | $7.72_{\pm0.33}$ | $7.33_{\pm0.13}$ | $47.61_{\pm0.83}$ | $43.05_{\pm0.54}$ |
| FlexMatch (Zhang et al., 2021a) | $26.76_{\pm1.12}$ | $18.24_{\pm0.36}$ | $14.40_{\pm3.11}$ | $8.17_{\pm0.78}$ | $7.82_{\pm0.77}$ | $7.41_{\pm0.38}$ | $45.73_{\pm1.60}$ | $42.25_{\pm0.33}$ |
| CoMatch (Li et al., 2021a) | $35.08_{\pm0.69}$ | $25.35_{\pm0.50}$ | $15.12_{\pm1.88}$ | $9.56_{\pm1.35}$ | $\underline{7.44}_{\pm0.30}$ | $7.72_{\pm1.14}$ | $48.76_{\pm0.90}$ | $43.36_{\pm0.21}$ |
| SimMatch (Zheng et al., 2022) | $23.78_{\pm1.08}$ | $17.06_{\pm0.78}$ | $\underline{11.77}_{\pm3.20}$ | $\mathbf{7.55_{\pm1.86}}$ | $7.93_{\pm0.55}$ | $\mathbf{7.08_{\pm0.33}}$ | $45.91_{\pm0.95}$ | $42.21_{\pm0.30}$ |
| FreeMatch (Wang et al., 2023) | $\mathbf{21.40_{\pm0.30}}$ | $\mathbf{15.65_{\pm0.26}}$ | $12.73_{\pm3.22}$ | $8.52_{\pm0.53}$ | $8.94_{\pm0.21}$ | $7.95_{\pm0.45}$ | $46.41_{\pm0.60}$ | $42.64_{\pm0.06}$ |
| SoftMatch (Chen et al., 2023) | $22.67_{\pm1.32}$ | $16.84_{\pm0.66}$ | $13.55_{\pm3.16}$ | $7.84_{\pm1.72}$ | $7.76_{\pm0.58}$ | $7.97_{\pm0.72}$ | $\underline{45.29}_{\pm0.95}$ | $\underline{42.21}_{\pm0.20}$ |
| Ours | $\underline{22.06}_{\pm1.06}$ | $\underline{16.40}_{\pm0.54}$ | $\mathbf{11.09_{\pm0.71}}$ | $8.10_{\pm1.02}$ | $\mathbf{7.32_{\pm0.12}}$ | $7.64_{\pm0.67}$ | $\mathbf{43.96_{\pm0.32}}$ | $42.32_{\pm0.02}$ |

methods with confidence thresholding, such as FixMatch (Sohn et al., 2020), AdaMatch (Berthelot et al., 2021), FlexMatch (Zhang et al., 2021a), FreeMatch (Wang et al., 2023), and SoftMatch (Chen et al., 2023). We also compare with methods with the contrastive loss, CoMatch (Li et al., 2021a) and SimMatch (Zheng et al., 2022). A full comparison of the USB datasets and hyper-parameters is shown in Appendix C.3.

**Results**. We present the results for SSL on Table 2. Although no individual SSL algorithm dominates the USB benchmark (Wang et al., 2022d), our method still shows competitive performance. Notably, our method performs best on STL-10 with 40 labels and Amazon Review with 250 labels, outperforming the previous best by **0.68**% and **1.33**%. In the other settings, the performance of our method is also very close to the best-performing methods. More remarkably, our method does not employ any thresholding, re-weighting, or contrastive techniques to achieve current results, demonstrating a significant potential to be further explored.

### 5.3 Noisy Label Learning

**Setup**. We conduct the experiments of NLL following SOP (Liu et al., 2022) on both synthetic symmetric/asymmetric noise on CIFAR-10 and CIFAR-100, and more realistic and larger-scale instance noise on Clothing1M (Xiao et al., 2015b), and WebVision (Li et al., 2017). To introduce the synthetic symmetric noise to CIFAR-10 and CIFAR-100, we uniformly flip labels for a probability $\eta$ into other classes. For asymmetric noise, we only randomly flip the labels for particular pairs of classes. We mainly select three previous best methods as baselines: DivideMix (Li et al., 2020); ELR (Liu et al., 2020); and SOP (Liu et al., 2022). We also include the normal cross-entropy (CE) training and mixup (Zhang et al., 2017) as baselines. More comparisons regarding other methods (Patrini et al., 2016; Han et al., 2018) and on CIFAR-10N (Wei et al., 2021) with training details and more baselines (Jiang et al., 2018; Han et al., 2018) are shown in Appendix C.4.

**Results**. We present the noisy label learning results in Table 3. The proposed method is comparable to the previous best methods. On synthetic noise of CIFAR-10, our method demonstrates the best performance on both symmetric noise and asymmetric noise. On CIFAR-100, our method generally produces similar results comparable to SOP. One may notice that our method shows inferior performance on asymmetric noise of CIFAR-100; we argue this is mainly due to the oversimplification of the noise transition model. Our method also achieves the best results on WebVision, outperforming the previous best by **2.05**%. On Clothing1M, our results are also very close to DivideMix, which trains for 80 epochs compared to 10 epochs in ours.

### 5.4 Mixed Imprecise Label Learning

**Setup**. We evaluate on CIFAR-10 and CIFAR-100 in a more challenging and realistic setting, the mixture of various imprecise label configurations, with unlabeled, partially labeled, and noisy labeled data existing simultaneously. We first sample the labeled dataset and treat other samples as the unlabeled. On the labeled dataset, we generate partial labels and randomly corrupt the true label of the partial labels. We set $l \in \{1000, 5000, 50000\}$ for CIFAR-10, and $l \in \{5000, 10000, 50000\}$ for CIFAR-100. For partial labels, we set $q \in \{0.1, 0.3, 0.5\}$ for CIFAR-10, and $q \in \{0.01, 0.05, 0.1\}$ for CIFAR-100. For noisy labels, we set $\eta \in \{0, 0.1, 0.2, 0.3\}$ for both datasets. Since there is no prior work that can handle all settings all at once, we compare on partial noisy label learning with PiCO+ (Wang et al., 2022b), IRNet (Lian et al., 2022b),

Table 3: Accuracy of synthetic noise on CIFAR-10 and CIFAR-100 and instance noise on Clothing1M and WebVision for **noisy label learning**. We use noise ratio of $\{0.2, 0.5, 0.8\}$ for synthetic symmetric noise and 0.4 for asymmetric label noise. The instance noise ratio is unknown.

| Dataset | CIFAR-10 | | | | CIFAR-100 | | | | Clothing1M | WebVision |
|---|---|---|---|---|---|---|---|---|---|---|
| Noise Type | Sym. | | | Asym. | Sym. | | | Asym. | Ins. | Ins. |
| Noise Ratio $\eta$ | 0.2 | 0.5 | 0.8 | 0.4 | 0.2 | 0.5 | 0.8 | 0.4 | - | - |
| CE | 87.20 | 80.70 | 65.80 | 82.20 | 58.10 | 47.10 | 23.80 | 43.30 | 69.10 | - |
| Mixup (Zhang et al., 2017) | 93.50 | 87.90 | 72.30 | - | 69.90 | 57.30 | 33.60 | - | - | - |
| DivideMix (Li et al., 2020) | 96.10 | 94.60 | 93.20 | 93.40 | 77.10 | 74.60 | 60.20 | 72.10 | **74.26** | 77.32 |
| ELR (Liu et al., 2020) | 95.80 | 94.80 | 93.30 | 93.00 | 77.70 | 73.80 | 60.80 | 77.50 | 72.90 | 76.20 |
| SOP (Liu et al., 2022) | 96.30 | 95.50 | 94.00 | 93.80 | **78.80** | **75.90** | 63.30 | **78.00** | 73.50 | 76.60 |
| Ours | **96.78**±0.11 | **96.60**±0.15 | **94.31**±0.07 | **94.75**±0.81 | 77.49±0.28 | 75.51±0.52 | **66.46**±0.72 | 75.82±1.89 | 74.02±0.12 | **79.37**±0.09 |

Table 4: Accuracy comparison of **mixture of different imprecise labels**. We report results of full labels, partial ratio $q$ of 0.1 (0.01) and 0.3 (0.05) for CIFAR-10 (CIFAR-100), and noise ratio $\eta$ of 0.1, 0.2, and 0.3 for CIFAR-10 and CIFAR-100.

| Method | $q$ | CIFAR-10, $l$=50000 | | | $q$ | CIFAR-100, $l$=50000 | | |
|---|---|---|---|---|---|---|---|---|
| | | $\eta$=0.1 | $\eta$=0.2 | $\eta$=0.3 | | $\eta$=0.1 | $\eta$=0.2 | $\eta$=0.3 |
| PiCO+ (Wang et al., 2022b) | 0.1 | 93.64 | 93.13 | 92.18 | 0.01 | 71.42 | 70.22 | 66.14 |
| IRNet (Lian et al., 2022b) | | 93.44 | 92.57 | 92.38 | | 71.17 | 70.10 | 68.77 |
| DALI (Xu et al., 2023) | | 94.15 | 94.04 | 93.77 | | 72.26 | 71.98 | 71.04 |
| PiCO+ Mixup (Xu et al., 2023) | | 94.58 | 94.74 | 94.43 | | 75.04 | 74.31 | 71.79 |
| DALI Mixup (Xu et al., 2023) | | 95.83 | 95.86 | 95.75 | | 76.52 | 76.55 | 76.09 |
| Ours | | **96.47**±0.11 | **96.09**±0.20 | **95.83**±0.05 | | **77.53**±0.24 | **76.96**±0.02 | **76.43**±0.27 |
| PiCO+ (Wang et al., 2022b) | 0.3 | 92.32 | 92.22 | 89.95 | 0.05 | 69.40 | 66.67 | 62.24 |
| IRNet (Lian et al., 2022b) | | 92.81 | 92.18 | 91.35 | | 70.73 | 69.33 | 68.09 |
| DALI (Xu et al., 2023) | | 93.44 | 93.25 | 92.42 | | 72.28 | 71.35 | 70.05 |
| PiCO+ Mixup (Xu et al., 2023) | | 94.02 | 94.03 | 92.94 | | 73.06 | 71.37 | 67.56 |
| DALI Mixup (Xu et al., 2023) | | 95.52 | 95.41 | 94.67 | | 76.87 | 75.23 | 74.49 |
| Ours | | **96.2**±0.02 | **95.87**±0.14 | **95.22**±0.06 | | **77.07**±0.16 | **76.34**±0.08 | **75.13**±0.63 |

Table 5: Robust test accuracy results of our method on **more mixture of imprecise label configurations**. $l$, $q$ and $\eta$ are the number of labels, partial, and noise ratio.

| $l$ | $q$ | CIFAR10 | | | | $l$ | $q$ | CIFAR100 | | | |
|---|---|---|---|---|---|---|---|---|---|---|---|
| | | $\eta$=0.0 | $\eta$=0.1 | $\eta$=0.2 | $\eta$=0.3 | | | $\eta$=0.0 | $\eta$=0.1 | $\eta$=0.2 | $\eta$=0.3 |
| 5,000 | 0.1 | 95.29±0.18 | 93.90±0.11 | 92.02±0.22 | 89.02±0.63 | 10,000 | 0.01 | 69.90±0.23 | 68.74±0.15 | 66.87±0.34 | 65.34±0.02 |
| | 0.3 | 95.13±0.16 | 92.95±0.37 | 90.14±0.61 | 87.31±0.27 | | 0.05 | 69.85±0.20 | 68.08±0.28 | 66.78±0.43 | 64.83±0.17 |
| | 0.5 | 95.04±0.10 | 92.18±0.52 | 88.39±0.62 | 83.09±0.56 | | 0.10 | 68.92±0.45 | 67.15±0.63 | 64.44±1.29 | 60.26±1.96 |
| 1,000 | 0.1 | 94.48±0.09 | 91.68±0.17 | 87.17±0.51 | 81.04±1.13 | 5,000 | 0.01 | 65.66±0.27 | 63.13±0.27 | 60.93±0.17 | 58.36±0.56 |
| | 0.3 | 94.35±0.05 | 89.94±1.90 | 82.06±1.52 | 69.20±2.16 | | 0.05 | 65.06±0.04 | 62.28±0.47 | 58.92±0.34 | 53.24±1.69 |
| | 0.5 | 93.92±0.29 | 86.34±2.37 | 70.86±2.78 | 38.19±6.55 | | 0.10 | 63.32±0.55 | 58.73±1.33 | 53.27±1.57 | 46.19±1.04 |

and DALI (Xu et al., 2023). Although there are also prior efforts on partial semi-supervised learning (Wang et al., 2019b; Wang & Zhang, 2020), they do not scale on simple dataset even on CIFAR-10. Thus, we did not include them in comparison. We conduct additional validation of our method on more complex settings for partial noisy labels with unlabeled data to demonstrate its robustness to various imprecise labels.

**Results**. We report the comparison with partial noisy label learning methods in Table 4. Compared to previous methods, the proposed method achieves the best performance. Despite the simplicity, our method outperforms PiCO+ and DALI with mixup, showing the effectiveness of dealing with mixed imprecise labels. We also report the results of our methods on more mixed imprecise label configurations in Table 5. Our method demonstrates significant robustness against various settings of the size of labeled data, partial ratio, and noise ratio. Note that this is the first work that naturally deals with all three imprecise label configurations simultaneously, with superior performance than previous methods handling specific types or combinations of label configurations. This indicates the enormous potential of our work in realistic applications for handling more practical and complicated data annotations common in real world applications.

## 6 Conclusion

We present the imprecise label learning (ILL) framework, a unified and consolidated solution for learning from all types of imprecise labels. ILL effectively employs an expectation-maximization (EM) algorithm for maximum likelihood estimation (MLE) of the distribution over the latent ground truth labels $Y$, imprecise label information $I$, and data $X$. It naturally extends and encompasses previous formulations for various imprecise label settings, achieving promising results. Notably, in scenarios where mixed configurations of imprecise labels coexist, our method exhibits substantial robustness against diverse forms of label imprecision. The potential **broader impact** of the ILL framework is substantial. It stands poised to transform domains where obtaining precise labels poses a challenge, offering a simple, unified, and effective approach to such contexts. Beyond the three imprecise label configurations we have demonstrated in this study, the ILL framework shows promise for an extension to more intricate scenarios such as multi-instance learning (Ilse et al., 2018) and multi-label crowd-sourcing learning (Ibrahim et al., 2023). However, it is also crucial to acknowledge the **limitations** of the ILL framework. Although its effectiveness has been substantiated on relatively smaller-scale datasets, additional empirical validation is necessary to assess its scalability to larger datasets. Furthermore, our study only considers balanced datasets; thus, the performance of the ILL framework when dealing with imbalanced data and open-set data still remains an open area for future exploration. We hope that our study will constitute a significant stride towards a comprehensive solution for imprecise label learning and catalyze further research in this crucial field.

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

# Appendix

## A Notation

We present the notation table for each symbol used in this paper in Table 6.

Table 6: Notation Table

| Notation | Definition |
|---|---|
| $\mathbf{x}$ | A training instance |
| $y$ | A class index label |
| $\{\mathbf{x}_i\}_{i \in [N]}$ | A set of data instances $\mathbf{x}$ of size $N$ |
| $\{y_i\}_{i \in [N]}$ | A set of precise label indices $y$ of size $N$ |
| $[\iota]$ | An imprecise label, which might contain multiple class indices |
| $\{[\iota]_i\}_{i \in [N]}$ | A set of imprecise labels $[\iota]$ of size $N$ |
| $X$ | Random variable of training instance |
| $\mathcal{X}$ | Input space where $\mathbf{x}$ is drawn from |
| $Y$ | Random variable of ground-truth labels |
| $\mathcal{Y}$ | Label space where $y$ is drawn from |
| $I$ | Random variable of imprecise labels |
| $f$ | Model backbone |
| $g$ | Model classifier |
| $h$ | Model multi-layer perceptron |
| $f \circ g$ | Model mapping $\mathcal{X} \to \mathcal{Y}$ |
| $\theta$ | Learnable parameters of $f \circ g$ |
| $\mathbf{p}(y\|\mathbf{x}; \theta)$ | Output probability from model $f \circ g$ |
| $f \circ h$ | Model mapping $\mathcal{X} \to \mathcal{Z}$, where $Z$ is a projected feature space |
| $\mathcal{D}$ | Dataset |
| $\mathcal{L}$ | Loss function |
| $\mathcal{A}_{\mathrm{w}}$ | Weak data augmentation, usually is HorizontalFlip |
| $\mathcal{A}_{\mathrm{s}}$ | Strong data augmentation, usually is RandAugment (Cubuk et al., 2020) |
| $\mathbf{z}_{\mathrm{w}}$ | Projected features from $f \circ h$ on weakly-augmented data |
| $\mathbf{z}_{\mathrm{s}}$ | Projected features from $f \circ h$ on strongly-augmented data |
| $\mathcal{M}$ | Memory queue in MoCo (He et al., 2020) |
| $\mathbf{s}$ | A partial label, with ground-truth label contained |
| $\{\mathbf{s}_i\}_{i \in [N]}$ | A set partial labels, with ground-truth label contained of size $N$ |
| $S$ | Random variable of partial label |
| $\mathbf{x}^{\mathrm{l}}$ | A labeled training example |
| $y^{\mathrm{l}}$ | A labeled class index |
| $\mathbf{x}^{\mathrm{u}}$ | A unlabeled training example |
| $y^{\mathrm{u}}$ | A unknown class index for unlabeled data |
| $X^{\mathrm{L}}$ | A set of labeled data instances |
| $Y^{\mathrm{L}}$ | A set of labels for labeled data instances |
| $X^{\mathrm{U}}$ | A set of unlabeled data instances |
| $Y^{\mathrm{U}}$ | A set of unknown labels for unlabeled data instances |
| $\hat{p}^{\mathbf{u}}$ | The maximum predicted probability on unlabeled data $\max(\mathbf{p}(y\|\mathbf{x}^{\mathrm{u}}; \theta))$ |
| $\hat{y}^{\mathrm{u}}$ | The pseudo-label from the predicted probability on unlabeled data $\arg\max(\mathbf{p}(y\|\mathbf{x}^{\mathrm{u}}; \theta))$ |
| $\tau$ | The threshold for confidence thresholding |
| $\hat{y}$ | A corrupted/noisy label |
| $\hat{y}^{\mathrm{oh}}$ | An one-hot version of the corrupted/noisy label |
| $\hat{Y}$ | Random variable of noisy labels |
| $\mathbf{u}, \mathbf{v}, \mathbf{m}$ | Noise model related parameters in SOP (Liu et al., 2022) |
| $\mathcal{T}(\hat{y}\|y; \omega)$ | The simplified noise transition model in ILL |
| $\omega$ | The parameters in the simplified noise model |

## B  Methods

### B.1  Derivation of Variational Lower Bound

Evidence lower bound (ELBO), or equivalently variational lower bound (Dempster et al., 1977), is the core quantity in EM. From Eq. (5), to model $\log P(X, I; \theta)$, we have:

$$
\begin{aligned}
\log P(X, I; \theta) &= \int Q(Y) \log P(X, I; \theta) dY \\
&= \int Q(Y) \log P(X, I; \theta) \frac{P(Y|X, I; \theta)}{P(Y|X, I; \theta)} dY \\
&= \int Q(Y) \log \frac{P(X, I, Y; \theta) Q(Y)}{P(Y|X, I; \theta) Q(Y)} dY \\
&= \int Q(Y) \log \frac{P(X, I, Y; \theta)}{Q(Y)} dY - \int Q(Y) \log \frac{P(Y|X, I; \theta)}{Q(Y)} dY
\end{aligned}
\tag{12}
$$

where the first term is the ELBO and the second term is the KL divergence $\mathcal{D}_{KL}(Q(Y)||P(Y|X, I; \theta))$. Replacing $Q(Y)$ with $P(Y|X, I; \theta^t)$ at each iteration will obtain Eq. (5).

### B.2  Instantiations to Partial Label Learning

The imprecise label $I$ for partial labels is defined as the label candidate sets $S$ with $\{\mathbf{s}_i\}_{i \in [N]}$ containing the true labels. Now we can derive Eq. (6) by replacing $I$ with $S$ in Eq. (5):

$$
\begin{aligned}
&\mathbb{E}_{Y|X, I; \theta^t} \left[ \log P(Y|X; \theta) + \log P(I|X, Y; \theta) \right] \\
&= \mathbb{E}_{Y|X, S; \theta^t} \left[ \log P(Y|X; \theta) + \log P(I|X, Y; \theta) \right] \\
&= \sum_Y P(Y|X, S; \theta^t) \left[ \log P(Y|X; \theta) + \log P(I|X, Y; \theta) \right] \\
&= \sum_Y P(Y|X, S; \theta^t) \left[ \log P(Y|X; \theta) \right] + \log P(I|X, Y; \theta)
\end{aligned}
\tag{13}
$$

Note that $P(I|Y, X; \theta)$ can be moved out of the expectation because it is a fixed quantity to any $Y$. Now we replace $Y$, $X$, and $S$ to $y$, $\mathbf{x}$, and $\mathbf{s}$ for each instance, and converting the maximization problem to negative log-likelihood minimization problem to drive the loss function:

$$
\mathcal{L}_{\text{ILL}}^{\text{PLL}} = -\frac{1}{N} \sum_i^N \mathbf{p}(y_i|\mathbf{x}_i, \mathbf{s}_i; \theta^t) \log \mathbf{p}(y_i|\mathbf{x}_i; \theta) - \frac{1}{N} \sum_i^N \log \mathbf{p}(\mathbf{s}_i|\mathbf{x}_i, y_i; \theta).
\tag{14}
$$

The first term is the Cross-Entropy loss we derived in Eq. (6). If $S$ is not instance-dependent, then knowing $Y$ also knows $S$, the second term thus can be ignored in Eq. (6). If $S$ becomes instance-dependent, the second term can be maintained as a supervised term as in (Wu et al., 2022) to optimize $\theta$.

### B.3  Instantiations to Semi-Supervised Learning

In SSL, the input $X$ consists of the labeled data $X^{\text{L}}$ and the unlabeled data $X^{\text{U}}$. The imprecise label for SSL is realized as the limited number of full labels $Y^{\text{L}}$ for $X^{\text{L}}$. The labels $Y^{\text{U}}$ for unlabeled $X^{\text{U}}$ are unknown and become the latent variable. Thus we can write:

$$
\begin{aligned}
&\mathbb{E}_{Y|X, I; \theta^t} \left[ \log P(Y|X; \theta) + \log P(I|X, Y; \theta) \right] \\
&= \mathbb{E}_{Y^{\text{U}}|X^{\text{U}}, X^{\text{L}}, Y^{\text{L}}; \theta^t} \left[ \log P(Y^{\text{U}}|X^{\text{U}}, X^{\text{L}}; \theta) + \log P(Y^{\text{L}}|X^{\text{L}}, X^{\text{U}}, Y^{\text{U}}; \theta) \right] \\
&= \sum_{Y^{\text{U}}} P(Y^{\text{U}}|X^{\text{U}}; \theta^t) \left[ \log P(Y^{\text{U}}|X^{\text{U}}; \theta) \right] + \log P(Y^{\text{L}}|X^{\text{L}}; \theta).
\end{aligned}
\tag{15}
$$

The negative log-likelihood loss function for $\{\mathbf{x}_i^l, y_i^l\}_{i \in [N^L]}$ and $\{\mathbf{x}^u\}_{i \in [N^U]}$ thus becomes:

$$
\mathcal{L}_{\text{ILL}}^{\text{SSL}} = \mathcal{L}_{\text{CE}} \left( \mathbf{p}(y|\mathbf{x}^{\text{u}}; \theta), \mathbf{p}(y|\mathbf{x}^{\text{u}}; \theta^t) \right) + \mathcal{L}_{\text{CE}} \left( \mathbf{p}(y|\mathbf{x}^{\text{L}}; \theta), y^{\text{L}} \right)
\tag{16}
$$

### B.4 Instantiations to Noisy Label Learning

We denote the given noisy labels as $\hat{Y}$. For noisy label learning, our method naturally supports a noise transition model $\mathcal{T}(\hat{Y}|Y;\omega)$ with learnable parameter $\omega$, as we will show in the following:

$$
\begin{aligned}
&\mathbb{E}_{Y|X,I;\theta^t}\left[\log P(Y|X;\theta) + \log P(I|X,Y;\theta)\right] \\
&= \mathbb{E}_{Y|X,\hat{Y};\theta^t}\left[\log P(Y,\hat{Y}|X;\theta)\right] \\
&= \mathbb{E}_{Y|X,\hat{Y};\theta^t}\left[\log P(Y|\hat{Y},X;\theta) + \log P(\hat{Y}|X;\theta)\right] \\
&= \sum_Y P(Y|\hat{Y},X;\theta^t)\log P(Y|\hat{Y},X;\theta) + \log P(\hat{Y}|X;\theta).
\end{aligned}
\tag{17}
$$

The loss function is:

$$
\mathcal{L}_{\mathrm{ILL}}^{\mathrm{NLL}} = \mathcal{L}_{\mathrm{CE}}\left(\mathbf{p}(y|\mathbf{x},\hat{y};\theta,\omega^t), \mathbf{p}(y|\mathbf{x},\hat{y};\theta^t,\omega^t)\right) + \mathcal{L}_{\mathrm{CE}}\left(\mathbf{p}(\hat{y}|\mathbf{x};\theta,\omega),\hat{y}\right)
\tag{18}
$$

Note that both term is computed from the noise transition matrix as mentioned in Eq. (9).

### B.5 Instantiations to Mixed Imprecise Label Learning

In this setting, we have both labeled data and unlabeled data, where the labels for the labeled data are both partial and noisy. On the unlabeled data, the unsupervised objective is the same as the unsupervised consistency regularization of semi-supervised learning shown in Eq. (7). On the labeled data, it mainly follows the Eq. (9) of noisy label learning, with the noisy single label becoming the noisy partial labels $\hat{\mathbf{s}}$. For noisy partial labels, the noisy supervised objective in Eq. 8 becomes the supervised consistency regularization as in Eq. 6 of partial label setting to train the noise transition model, and the noisy unsupervised objective becomes the consistency regularization of the prediction conditioned on noisy partial labels:

$$
\mathcal{L}_{\mathrm{CE}}\left(\mathbf{p}\left(y \mid \mathcal{A}_{\mathrm{s}}(\mathbf{x}),\hat{\mathbf{s}};\theta,\omega^t\right), \mathbf{p}\left(y \mid \mathcal{A}_{\mathrm{w}}(\mathbf{x}),\hat{y};\theta^t,\omega^t\right)\right) + \mathcal{L}_{\mathrm{CE}}\left(\mathbf{p}\left(\hat{y} \mid \mathcal{A}_{\mathrm{w}}(\mathbf{x});\theta,\omega\right),\hat{\mathbf{s}}\right)
\tag{19}
$$

We can compute both quantity through the noise transition model:

$$
\mathbf{p}(y|\mathbf{x},\hat{\mathbf{s}};\theta,\omega^t) \propto \mathbf{p}(y|\mathbf{x};\theta)\prod_{\hat{y}\in\hat{\mathbf{s}}}\mathcal{T}(y|\hat{y};\omega^t), \text{and } \mathbf{p}(\hat{y}|\mathbf{x};\theta,\omega) = \sum_{y\in[C]}\mathbf{p}(y|\mathbf{x};\theta)\mathcal{T}(\hat{y}|y;\omega).
\tag{20}
$$

## C Experiments

### C.1 Additional Training Details

We adopt two additional training strategies for the ILL framework. The first is the "strong-weak" augmentation strategy (Xie et al., 2020a). Since there is a consistency regularization term in each imprecise label formulation of ILL, we use the soft pseudo-targets of the weakly-augmented data to train the strongly-augmented data. The second is the entropy loss (John Bridle, 1991) for class balancing, which is also adopted in SOP (Liu et al., 2022) and FreeMatch (Wang et al., 2023). We set the loss weight for the entropy loss uniformly for all experiments as 0.1.

### C.2 Partial Label Learning

#### C.2.1 Setup

Following previous work (Xu et al., 2022; Wen et al., 2021; Wang et al., 2022a), we evaluate our method on partial label learning setting using CIFAR-10, CIFAR-100, and CUB-200 (Welinder et al., 2010). We generate partially labeled datasets by flipping negative labels to false positive labels with a probability $q$, which is also denoted as a partial ratio. Specifically, the $C-1$ negative labels are uniformly aggregated into the ground

truth label to form a set of label candidates. We consider $q \in \{0.1, 0.3, 0.5\}$ for CIFAR-10, $q \in \{0.01, 0.05, 0.1\}$ for CIFAR-100, and $q = 0.05$ for CUB-200. For CIFAR-10 and CIFAR-100, we use ResNet-18 (He et al., 2016) as backbone. We use SGD as an optimizer with a learning rate of 0.01, a momentum of 0.9, and a weight decay of $1e-3$. For CUB-200, we initialize the ResNet-18 (He et al., 2016) with ImageNet-1K (Deng et al., 2009) pre-trained weights. We train 800 epochs for CIFAR-10 and CIFAR-100 (Krizhevsky et al., 2009), and 300 epochs for CUB-200, with a cosine learning rate scheduler. For CIFAR-10 and CIFAR-100, we use an input image size of 32. For CUB-200, we use an input image size of 224. A batch size of 256 is used for all datasets. The choice of these parameters mainly follows PiCO (Wang et al., 2022a). We present the full hyper-parameters systematically in Table 7.

Table 7: Hyper-parameters for **partial label learning** used in experiments.

| Hyper-parameter | CIFAR-10 | CIFAR-100 | CUB-200 |
|---|---|---|---|
| Image Size | 32 | 32 | 224 |
| Model | ResNet-18 | ResNet-18 | ResNet-18 (ImageNet-1K Pretrained) |
| Batch Size | 256 | 256 | 256 |
| Learning Rate | 0.01 | 0.01 | 0.01 |
| Weight Decay | 1e-3 | 1e-3 | 1e-5 |
| LR Scheduler | Cosine | Cosine | Cosine |
| Training Epochs | 800 | 800 | 300 |
| Classes | 10 | 100 | 200 |

### C.2.2 Discussion

We additionally compare our method with R-CR (Wu et al., 2022), which uses a different architecture as the results in Table 1. R-CR uses Wide-ResNet34x10 as backbone, and adopts multiple strong data augmentations. It also adjusts the loss weight along training. For fair comparison, we use the same architecture without multiple augmentation and the curriculum adjust on loss. The results are shown in Table 8, where our method outperforms R-CR on CIFAR-10 and is comparable on CIFAR-100.

Table 8: Comparison with R-CR in partial label learning

| Method | CIFAR-10 | | CIFAR-100 | |
|---|---|---|---|---|
| | 0.3 | 0.5 | 0.05 | 0.10 |
| R-CR | $97.28_{\pm 0.02}$ | $97.05_{\pm 0.05}$ | $82.77_{\pm 0.10}$ | $82.24_{\pm 0.07}$ |
| Ours | $97.55_{\pm 0.07}$ | $97.17_{\pm 0.11}$ | $82.46_{\pm 0.08}$ | $82.22_{\pm 0.05}$ |

We also provide the comparison of our method on instance-dependent partial label learning as proposed in Xu et al. (2021a; 2022). Due to the nature of instance-dependence, we maintain the term $P(S|Y, X; \theta)$ from Eq. (5) as a supervised term for optimization. We compare our method with VALEN (Xu et al., 2021a), RCR (Wu et al., 2022), PiCO Wang et al. (2022a), and POP (Xu et al., 2022) on MNIST, Kuzushiji-MNIST, Fashion-MNIST, CIFAR-10, and CIFAR-100, with synthetic instance-dependent partial labels generated according to Xu et al. (2022). From the results in Table 9, we proposed method demonstrate the best performance across different datasets evaluated.

Table 9: Comparison on instance-dependent partial label learning

| | MNIST | Kuzushiji-MNIST | Fashion-MNIST | CIFAR-10 | CIFAR-100 |
|---|---|---|---|---|---|
| VALEN (Xu et al., 2021a) | 99.03 | 90.15 | 96.31 | 92.01 | 71.48 |
| RCR (Wu et al., 2022) | 98.81 | 90.62 | 96.64 | 86.11 | 71.07 |
| PiCO (Wang et al., 2022a) | 98.76 | 88.87 | 94.83 | 89.35 | 66.30 |
| POP (Xu et al., 2022) | **99.28** | 91.09 | 96.93 | 93.00 | 71.82 |
| Ours | 99.19 | **91.35** | **97.01** | **93.86** | **72.43** |

A recent work on PLL discussed and analyzed the robustness performance of different loss functions, especially the average-based methods (Lv et al., 2023). We perform a similar analysis here for the derived loss function in ILL. Following the notation in Lv et al. (2023), let $\mathbf{s}$ denote the candidate label set, $\mathbf{x}$ as the training instance, $g$ as the probability score from the model, and $f$ as the classifier $f(\boldsymbol{x}) = \arg\max_{i \in \mathcal{Y}} g_i(\boldsymbol{x})$, the average-based PLL can be formulated as:

$$\mathcal{L}_{avg-PLL}(f(\boldsymbol{x}), \mathbf{s}) = \frac{1}{|\mathbf{s}|} \sum_{i \in \mathbf{s}} \ell(f(\boldsymbol{x}), i) \tag{21}$$

Lv et al. (2023) compared different loss functions $\ell$ on both noise-free and noisy PLL settings, where they find both theoretically and empirically that average-based PLL with *bounded* loss are robust under mild assumptions. Empirical study in Lv et al. (2023) suggests that both *Mean Absolute Error* and *Generalized Cross-Entropy* loss (Zhang & Sabuncu, 2018) that proposed for noisy label learning achieves the best performance and robustness for average-based PLL.

Our solution for PLL can be viewed as an instantiation of the average-based PLL as in (Lv et al., 2023) with:

$$\ell(f(\mathbf{x}), i) = -\bar{g}_i(\mathbf{x}) \log g_i(\mathbf{x}) \tag{22}$$

where $\bar{g}$ is normalized probability over $\mathbf{s}$ with detached gradient. We can further show that the above loss function is bounded for $0 < \ell \leq \frac{1}{e}$ and thus bounded for summation of all classes, which demonstrates robustness, as we show in Table 4.

### C.3 Semi-Supervised Learning

### C.3.1 Setup

For experiments of SSL, we follow the training and evaluation protocols of USB (Wang et al., 2022d) on image and text classification. To construct the labeled dataset for semi-supervised learning, we uniformly select $l/C$ samples from each class and treat the remaining samples as the unlabeled dataset. For image classification tasks, ImageNet-1K (Deng et al., 2009) Vision Transformers (Dosovitskiy et al., 2020) are used, including CIFAR-100 (Krizhevsky et al., 2009), EuroSAT (Helber et al., 2019), STL-10 (Coates et al., 2011), TissueMNIST (Yang et al., 2021a;b), Semi-Aves (Su & Maji, 2021). For text classification tasks, we adopt BERT (Devlin et al., 2018) as backbone, including IMDB (Maas et al., 2011), Amazon Review (McAuley & Leskovec, 2013), Yelp Review (yel), AG News (Zhang et al., 2015) , Yahoo Answer (Chang et al., 2008). The hyper-parameters strictly follow USB, and are shown in Table 10 and Table 11.

Table 10: Hyper-parameters of **semi-supervised learning** used in vision experiments of USB.

| Hyper-parameter | CIFAR-100 | STL-10 | Euro-SAT | TissueMNIST | Semi-Aves |
|---|---|---|---|---|---|
| Image Size | 32 | 96 | 32 | 32 | 224 |
| Model | ViT-S-P4-32 | ViT-B-P16-96 | ViT-S-P4-32 | ViT-T-P4-32 | ViT-S-P16-224 |
| Labeled Batch size | | | 16 | | |
| Unlabeled Batch size | | | 16 | | |
| Learning Rate | 5e-4 | 1e-4 | 5e-5 | 5e-5 | 1e-3 |
| Weight Decay | | | 5e-4 | | |
| Layer Decay Rate | 0.5 | 0.95 | 1.0 | 0.95 | 0.65 |
| LR Scheduler | | | $\eta = \eta_0 \cos(\frac{7\pi k}{16K})$ | | |
| Training epochs | | | 20 | | |
| Classes | 100 | 10 | 10 | 10 | 200 |
| Model EMA Momentum | | | 0.0 | | |
| Prediction EMA Momentum | | | 0.999 | | |
| Weak Augmentation | | Random Crop, Random Horizontal Flip | | | |
| Strong Augmentation | | RandAugment (Cubuk et al., 2020) | | | |

### C.3.2 Results

In the main paper, we only provide the comparison on CIFAR-100, STL-10, IMDB, and Amazon Review. Here we provide the full comparison in Table 12 and Table 13. From the full results, similar conclusion can be drawn as in the main paper. Our ILL framework demonstrates comparable performance as previous methods.

Table 11: Hyper-parameters of **semi-supervised learning** NLP experiments in USB.

| Hyper-parameter | AG News | Yahoo! Answer | IMDB | Amazom-5 | Yelp-5 |
|---|---|---|---|---|---|
| Max Length | | | 512 | | |
| Model | | | Bert-Base | | |
| Labeled Batch size | | | 4 | | |
| Unlabeled Batch size | | | 4 | | |
| Learning Rate | 5e-5 | 1e-4 | 5e-5 | 1e-5 | 5e-5 |
| Weight Decay | | | 1e-4 | | |
| Layer Decay Rate | 0.65 | 0.65 | 0.75 | 0.75 | 0.75 |
| LR Scheduler | | | $\eta = \eta_0 \cos(\frac{7\pi k}{16K})$ | | |
| Training epochs | | | 10 | | |
| Classes | 4 | 10 | 2 | 5 | 5 |
| Model EMA Momentum | | | 0.0 | | |
| Prediction EMA Momentum | | | 0.999 | | |
| Weak Augmentation | | | None | | |
| Strong Augmentation | | | Back-Translation (Xie et al., 2020a) | | |

Table 12: Error rate comparison of different number of labels on CIFAR-100, STL-10, EuroSAT, TissueMNIST, and SemiAves for **semi-supervised learning**. We use USB (Wang et al., 2022d) image classification task results. The best results are indicated in bold. Our results are averaged over 3 independent runs.

| Datasets | CIFAR-100 | | STL-10 | | EuroSat | | TissueMNIST | | SemiAves |
|---|---|---|---|---|---|---|---|---|---|
| # Labels | 200 | 400 | 40 | 100 | 20 | 40 | 80 | 400 | 3959 |
| Pseudo-Label (Lee et al., 2013) | 33.99±0.95 | 25.32±0.29 | 19.14±1.33 | 10.77±0.60 | 25.46±1.36 | 15.70±2.12 | 56.92±4.54 | 50.86±1.79 | 40.35±0.30 |
| Mean-Teacher (Tarvainen & Valpola, 2017) | 35.47±0.40 | 26.03±0.30 | 18.67±1.69 | 24.19±10.15 | 26.83±1.46 | 15.85±1.66 | 62.06±3.43 | 55.12±2.53 | 38.55±0.21 |
| VAT (Miyato et al., 2018) | 31.49±1.33 | 21.34±0.50 | 18.45±1.47 | 10.69±0.51 | 26.16±0.96 | 10.09±0.94 | 57.49±5.47 | 51.30±1.73 | 38.82±0.04 |
| MixMatch (Berthelot et al., 2019b) | 38.22±0.71 | 26.72±0.72 | 58.77±1.98 | 36.74±1.24 | 24.85±4.85 | 17.28±2.67 | 55.53±1.51 | 49.64±2.28 | 37.25±0.08 |
| ReMixMatch (Berthelot et al., 2019a) | 22.21±2.21 | 16.86±0.57 | 13.08±3.34 | **7.21±0.39** | **5.05±1.05** | 5.07±0.56 | 58.77±4.43 | 49.82±1.18 | **30.20±0.03** |
| AdaMatch (Berthelot et al., 2021) | 22.32±1.73 | 16.66±0.62 | 13.64±2.49 | 7.62±1.90 | 7.02±0.79 | **4.75±1.10** | 58.35±4.87 | 52.40±2.08 | 31.75±0.13 |
| FixMatch (Sohn et al., 2020) | 29.60±0.90 | 19.56±0.52 | 16.15±1.89 | 8.11±0.68 | 13.44±3.53 | 5.91±2.02 | **55.37±4.50** | 51.24±1.56 | 31.90±0.06 |
| FlexMatch (Zhang et al., 2021a) | 26.76±1.12 | 18.24±0.36 | 14.40±3.11 | 8.17±0.78 | 5.17±0.57 | 5.58±0.81 | 58.36±3.80 | 51.89±3.21 | 32.48±0.15 |
| Dash (Xu et al., 2021b) | 30.61±0.98 | 19.38±0.10 | 16.22±5.95 | 7.85±0.74 | 11.19±0.90 | 6.96±0.87 | 56.98±2.93 | 51.97±1.55 | 32.38±0.16 |
| CoMatch (Li et al., 2021a) | 35.08±0.69 | 25.35±0.50 | 15.12±1.88 | 9.56±1.35 | 5.75±0.43 | 4.81±1.05 | 59.04±4.90 | 52.92±1.04 | 38.65±0.18 |
| SimMatch (Zheng et al., 2022) | 23.78±1.08 | 17.06±0.78 | 11.77±3.20 | 7.55±1.86 | 7.66±0.60 | 5.27±0.89 | 60.88±4.31 | 52.93±1.56 | 33.85±0.08 |
| FreeMatch (Wang et al., 2023) | **21.40±0.30** | **15.65±0.26** | 12.73±3.22 | 8.52±0.53 | 6.50±0.78 | 5.78±0.51 | 58.24±3.08 | 52.19±1.35 | 32.85±0.31 |
| SoftMatch (Chen et al., 2023) | 22.67±1.32 | 16.84±0.66 | 13.55±3.16 | 7.84±1.72 | 5.75±0.62 | 5.90±1.42 | 57.98±3.66 | 51.73±2.84 | 31.80±0.22 |
| Ours | 22.06±1.06 | 17.40±1.04 | **11.09±0.71** | 8.10±1.02 | 5.86±1.06 | 5.74±1.13 | 57.99±2.16 | **50.95±2.03** | 33.08±0.26 |

Table 13: Error rate comparison of different number of labels on IMDB, AG News, Amazon Review, Yahoo Answers, and Yelp Review for **semi-supervised learning**. We use USB (Wang et al., 2022d) text classification task results. Best results are indicated in bold. Our results are averaged over 3 independent runs.

| Datasets | IMDB | | AG News | | Amazon Review | | Yahoo Answers | | Yelp Review | |
|---|---|---|---|---|---|---|---|---|---|---|
| # Labels | 20 | 100 | 40 | 200 | 250 | 1000 | 500 | 2000 | 250 | 1000 |
| Pseudo-Label (Lee et al., 2013) | 45.45±4.43 | 19.67±1.01 | 19.49±3.07 | 14.69±1.88 | 53.45±1.9 | 47.00±0.79 | 37.70±0.65 | 32.72±0.31 | 54.51±0.82 | 47.33±0.20 |
| Mean-Teacher (Tarvainen & Valpola, 2017) | 20.06±2.51 | 13.97±1.49 | 15.17±1.21 | 13.93±0.65 | 52.14±0.52 | 47.66±0.84 | 37.09±0.18 | 33.43±0.28 | 50.60±0.62 | 47.21±0.31 |
| VAT (Miyato et al., 2018) | 25.93±2.58 | 11.61±1.79 | 14.70±1.19 | 11.71±0.84 | 49.83±0.46 | 46.54±0.31 | 34.87±0.41 | 31.50±0.35 | 52.97±1.41 | 45.30±0.32 |
| MixMatch (Berthelot et al., 2019b) | 26.12±6.13 | 15.47±0.65 | 13.50±1.51 | 11.75±0.60 | 59.54±0.67 | 61.69±3.32 | 35.75±0.71 | 33.62±0.14 | 53.98±0.59 | 51.70±0.68 |
| AdaMatch (Berthelot et al., 2021) | 8.09±0.99 | 7.11±0.20 | **11.73±0.17** | **11.22±0.95** | 46.72±0.72 | 42.27±0.25 | 32.75±0.35 | 30.44±0.31 | 45.40±0.96 | 40.16±0.49 |
| FixMatch (Sohn et al., 2020) | 7.72±0.33 | 7.33±0.13 | 30.17±1.87 | 11.71±1.95 | 47.61±0.83 | 43.05±0.54 | **33.03±0.49** | 30.51±0.53 | 46.52±0.94 | 40.65±0.46 |
| FlexMatch (Zhang et al., 2021a) | 7.82±0.77 | 7.41±0.38 | 16.38±3.94 | 12.08±0.73 | 45.73±1.60 | 42.25±0.33 | 35.61±1.08 | 31.13±0.18 | **43.35±0.69** | 40.51±0.34 |
| Dash (Xu et al., 2021b) | 8.34±0.86 | 7.55±0.35 | 17.67±3.19 | 13.76±1.67 | 47.10±0.74 | 43.09±0.60 | 35.26±0.33 | 31.19±0.29 | 45.24±2.02 | 40.14±0.79 |
| CoMatch (Li et al., 2021a) | 7.44±0.30 | 7.72±1.14 | 11.95±0.76 | 10.75±0.35 | 48.76±0.90 | 43.36±0.21 | 33.48±0.51 | 30.25±0.35 | 45.40±1.12 | 40.27±0.51 |
| SimMatch (Zheng et al., 2022) | 7.93±0.55 | **7.08±0.33** | 14.26±1.51 | 12.45±1.37 | 45.91±0.95 | 42.21±0.30 | 33.06±0.20 | 30.16±0.21 | 46.12±0.48 | 40.26±0.62 |
| FreeMatch (Wang et al., 2023) | 8.94±0.21 | 7.95±0.45 | 12.98±0.98 | 11.73±0.63 | 46.41±0.60 | 42.64±0.06 | 32.77±0.26 | 30.32±0.18 | 47.95±1.45 | 40.37±1.00 |
| SoftMatch (Chen et al., 2023) | 7.76±0.58 | 7.97±0.72 | 11.90±0.27 | 11.72±1.58 | 45.29±0.95 | **42.21±0.20** | 33.07±0.31 | 30.44±0.62 | 44.09±0.50 | 39.76±0.13 |
| Ours | **7.32±0.12** | 7.64±0.67 | 14.77±1.59 | 12.21±0.82 | **43.96±0.32** | 42.32±0.02 | 33.80±0.25 | 30.86±0.17 | 44.82±0.17 | **39.67±0.71** |

## C.4 Noisy Label Learning

### C.4.1 Setup

We conduct experiments of noisy label learning following SOP (Liu et al., 2022). We evaluate the proposed method on both synthetic symmetric/asymmetric noise on CIFAR-10 and CIFAR-100, and more realistic and

larger-scale instance noise on Clothing1M and WebVision. To introduce the synthetic symmetric noise to CIFAR-10 and CIFAR-100, we uniformly flip labels for a probability $\eta$ into other classes. For asymmetric noise, we only randomly flip the labels for particular pairs of classes. For CIFAR-10 and CIFAR-100, we train PreAct-ResNet-18 with SGD using a learning rate of 0.02, a weight decay of $1e-3$, and a momentum of 0.9. We train for 300 epochs with a cosine learning rate schedule and a batch size of 128. For WebVision, we use InceptionResNet-v2 as the backbone and set the batch size to 32. Other settings are similar to CIFAR-10. For Clothing1M, we use ImageNet-1K pre trained ResNet-50 as the backbone. We train it using SGD with an initial learning rate of $2e$-3 for a total of 10 epochs, where the learning rate is reduced by 10 after 5 epochs. In addition, we also conduct experiments on CIFAR-10N and CIFAR-100N. We present the detailed hyper-parameters in Table 14.

Table 14: Hyper-parameters for **noisy label learning** used in experiments.

| Hyper-parameter | CIFAR-10 (CIFAR-10N) | CIFAR-100 (CIFAR-100N) | Clothing1M | WebVision |
|---|---|---|---|---|
| Image Size | 32 | 32 | 224 | 299 |
| Model | PreAct-ResNet-18 (ResNet-34) | PreAct-ResNet-18 (ResNet-34) | ResNet-50 (ImageNet-1K Pretrained) | Inception-ResNet-v2 |
| Batch Size | 128 | 128 | 64 | 32 |
| Learning Rate | 0.02 | 0.02 | 0.002 | 0.02 |
| Weight Decay | 1e-3 | 1e-3 | 1e-3 | 5e-4 |
| LR Scheduler | Cosine | Cosine | MultiStep | MultiStep |
| Training Epochs | 300 | 300 | 10 | 100 |
| Classes | 10 | 100 | 14 | 50 |
| Noisy Matrix Scale | 1.0 | 2.0 | 0.5 | 2.5 |

### C.4.2 Results

In addition to the results regarding noisy label learning provided in the main paper, we also present comparison results on CIFAR-10N and CIFAR-100N (Wei et al., 2021) in Table 15. We include a full comparison on Clothing1M and WebVision, incorporating methods like Co-Teaching, Forward, and CORES, in Table 16. As shown in Table 15, the proposed ILL framework achieves performance comparable to the previous best method, SOP (Liu et al., 2022). On CIFAR-10N, our method yields results very close to SOP in the Random and Aggregate case noise scenarios and surpasses SOP in the Worst case noise scenario. However, on CIFAR-100N, our method slightly underperforms previous methods, possibly due to the oversimplified noise model utilized in ILL. We believe that a more realistic noise transition model and further tuning of our method could lead to improved performance.

Table 15: Test accuracy comparison of instance independent label noise on CIFAR-10N and CIFAR-100N for **noisy label learning**. The best results are indicated in **bold**, and the second best results are indicated in underline. Our results are averaged over three independent runs with ResNet34 as the backbone.

| Dataset | CIFAR-10N | | | | | | CIFAR-100N | |
|---|---|---|---|---|---|---|---|---|
| Noisy Type | Clean | Random 1 | Random 2 | Random 3 | Aggregate | Worst | Clean | Noisy |
| CE | $92.92_{\pm0.11}$ | $85.02_{\pm0.65}$ | $86.46_{\pm1.79}$ | $85.16_{\pm0.61}$ | $87.77_{\pm0.38}$ | $77.69_{\pm1.55}$ | $76.70_{\pm0.74}$ | $55.50_{\pm0.66}$ |
| Forward (Patrini et al., 2016) | $93.02_{\pm0.12}$ | $86.88_{\pm0.50}$ | $86.14_{\pm0.24}$ | $87.04_{\pm0.35}$ | $88.24_{\pm0.22}$ | $79.79_{\pm0.46}$ | $76.18_{\pm0.37}$ | $57.01_{\pm1.03}$ |
| Co-teaching (Han et al., 2018) | $93.35_{\pm0.14}$ | $90.33_{\pm0.13}$ | $90.30_{\pm0.17}$ | $90.15_{\pm0.18}$ | $91.20_{\pm0.13}$ | $83.83_{\pm0.13}$ | $73.46_{\pm0.09}$ | $60.37_{\pm0.27}$ |
| DivideMix (Li et al., 2020) | - | $\underline{95.16_{\pm0.19}}$ | $\underline{95.23_{\pm0.07}}$ | $\underline{95.21_{\pm0.14}}$ | $95.01_{\pm0.71}$ | $92.56_{\pm0.42}$ | - | $71.13_{\pm0.48}$ |
| ELR (Liu et al., 2020) | $95.39_{\pm0.05}$ | $94.43_{\pm0.41}$ | $94.20_{\pm0.24}$ | $94.34_{\pm0.22}$ | $94.83_{\pm0.10}$ | $91.09_{\pm1.60}$ | $\underline{78.57_{\pm0.12}}$ | $66.72_{\pm0.07}$ |
| CORES (Cheng et al., 2020) | $94.16_{\pm0.11}$ | $94.45_{\pm0.14}$ | $94.88_{\pm0.31}$ | $94.74_{\pm0.03}$ | $95.25_{\pm0.09}$ | $91.66_{\pm0.09}$ | $73.87_{\pm0.16}$ | $55.72_{\pm0.42}$ |
| SOP (Liu et al., 2022) | $\mathbf{96.38_{\pm0.31}}$ | $\underline{95.28_{\pm0.13}}$ | $95.31_{\pm0.10}$ | $95.39_{\pm0.11}$ | $95.61_{\pm0.13}$ | $\underline{93.24_{\pm0.21}}$ | $\mathbf{78.91_{\pm0.43}}$ | $\underline{67.81_{\pm0.23}}$ |
| Ours | $\underline{96.21_{\pm0.29}}$ | $\mathbf{96.06_{\pm0.07}}$ | $\mathbf{95.98_{\pm0.12}}$ | $\mathbf{96.10_{\pm0.05}}$ | $\mathbf{96.40_{\pm0.03}}$ | $\mathbf{93.55_{\pm0.14}}$ | $78.53_{\pm0.21}$ | $\mathbf{68.07_{\pm0.33}}$ |

## C.5 Mixed Imprecise Label Learning

### C.5.1 Setup

To create a mixture of various imprecise label configurations, we select CIFAR-10 and CIFAR-100 as base datasets. We first uniformly sample $l/C$ labeled samples from each class to form the labeled dataset and treat the remaining samples as the unlabeled dataset. Based on the labeled dataset, we generate partially

Table 16: Test accuracy comparison of realistic noisy labels on Clothing1M and WebVision for **noisy label learning**. The best results are indicated in **bold** and the second best results are indicated in underline. Our results are averaged over 3 independent runs. For Clothing1M, we use ImageNet-1K pre trained ResNet50 as the backbone. For WebVision, InceptionResNetv2 is used as the backbone.

| Dataset | Clothing1M | WebVision |
|---|---|---|
| CE | 69.10 | - |
| Forward (Patrini et al., 2016) | 69.80 | 61.10 |
| MentorNet (Jiang et al., 2018) | 66.17 | 63.00 |
| Co-Teaching (Han et al., 2018) | 69.20 | 63.60 |
| DivideMix (Li et al., 2020) | **74.76** | 77.32 |
| ELR (Liu et al., 2020) | 72.90 | 76.20 |
| CORES (Cheng et al., 2020) | 73.20 | - |
| SOP (Liu et al., 2022) | 73.50 | 76.60 |
| Ours | 74.02$_{\pm 0.12}$ | **79.37$_{\pm 0.09}$** |

labeled datasets by flipping negative labels to false positive labels with the partial ratio $q$. After obtaining the partial labels, we randomly select $\eta$ percentage of samples from each class, and recreate the partial labels for them by flipping the ground truth label uniformly to another class. In this setting, unlabeled data, partially labeled data, and noisy labeled data exist simultaneously, which is very challenging and more closely resembles realistic situations. For CIFAR-10, we set $l \in \{1000, 5000, 50000\}$, and for CIFAR-100, we set $l \in \{5000, 10000, 50000\}$. Similarly in the partial label setting, we set $q \in \{0.1, 0.3, 0.5\}$ for CIFAR-10, and $q \in \{0.01, 0.05, 0.1\}$ for CIFAR-100. For noisy labels, we set $\eta \in \{0.1, 0.2, 0.3\}$ for both datasets.

### C.5.2    Results

We provide a more complete version of Table 4 in Table 17. On partial noisy labels of CIFAR-10 with partial ratio 0.5 and of CIFAR-100 with partial ratio 0.1, most baseline methods are more robust or even fail to perform. However, our ILL still shows very robust performance with minor performance degradation as increase of noise ratios.

### C.6    Ablation on Strong-Augmentation and Entropy Loss

We provide the ablation study on the strong-augmentation and entropy loss components here, which are common techniques in each setting (Sohn et al., 2020; Wang et al., 2022a; Liu et al., 2022). For example, in SSL, strong-weak augmentation is an important strategy for SSL algorithms widely used in existing works such as FixMatch (Sohn et al., 2020) and FlexMatch (Zhang et al., 2021a). Thus, it is important to adopt strong-weak augmentation to achieve better performance in SSL (Wang et al., 2023; Chen et al., 2023; Wang et al., 2022d). This is similar in PLL settings (Wang et al., 2022a; Wu et al., 2022). PiCO (Wang et al., 2022a; Wu et al., 2022) also used strong augmentation). Strong-weak augmentation and entropy loss are also adopted in SOP (Liu et al., 2022) of NLL. However, we found these techniques are less important for our formulation of NLL. We provide an ablation study on the entropy loss of SSL, and both techniques for NLL and PLL here to demonstrate our discussions.

### C.7    Runtime Analysis

We provide the runtime analysis on CIFAR-100 of our method on different settings, compared with the SOTA baselines. We compute the average runtime from all training iterations on CIFAR-100. The results are shown in Table 21. Since the settings studied in this work has loss functions derived as close-form from Eq. (5), the time complexity can be viewed as $\mathcal{O}(1)$. Thus our method in general present faster runtime without complex design such as contrastive loss.

Table 17: Test accuracy comparison of **mixture of different imprecise labels**. We report results of full labels, partial ratio $q$ of $\{0.1, 0.3, 0.5\}$ for CIFAR-10 and $\{0.01, 0.05, 0.1\}$ for CIFAR-100, and noise ratio $\eta$ of $\{0.1, 0.2, 0.3\}$ for CIFAR-10 and CIFAR-100.

| Dataset | # Labels | Partial Ratio $q$ | Noise Ratio $\eta$ | 0 | 0.1 | 0.2 | 0.3 |
|---|---|---|---|---|---|---|---|
| CIFAR-10 | 50,000 | 0.1 | PiCO+ (Wang et al., 2022b) | $95.99_{\pm0.03}$ | 93.64 | 93.13 | 92.18 |
| | | | IRNet (Lian et al., 2022b) | - | 93.44 | 92.57 | 92.38 |
| | | | DALI (Xu et al., 2023) | - | 94.15 | 94.04 | 93.77 |
| | | | PiCO+ w/ Mixup (Xu et al., 2023) | - | 94.58 | 94.74 | 94.43 |
| | | | DALI w/ Mixup (Xu et al., 2023) | - | 95.83 | 95.86 | 95.75 |
| | | | Ours | $\mathbf{96.55_{\pm0.08}}$ | $\mathbf{96.47_{\pm0.11}}$ | $\mathbf{96.09_{\pm0.20}}$ | $\mathbf{95.83_{\pm0.05}}$ |
| | | 0.3 | PiCO+ (Wang et al., 2022b) | $95.73_{\pm0.10}$ | 92.32 | 92.22 | 89.95 |
| | | | IRNet (Lian et al., 2022b) | - | 92.81 | 92.18 | 91.35 |
| | | | DALI (Xu et al., 2023) | - | 93.44 | 93.25 | 92.42 |
| | | | PiCO+ w/ Mixup (Xu et al., 2023) | - | 94.02 | 94.03 | 92.94 |
| | | | DALI w/ Mixup (Xu et al., 2023) | - | 95.52 | 95.41 | 94.67 |
| | | | Ours | $\mathbf{96.52_{\pm0.12}}$ | $\mathbf{96.2_{\pm0.02}}$ | $\mathbf{95.87_{\pm0.14}}$ | $\mathbf{95.22_{\pm0.06}}$ |
| | | 0.5 | PiCO+ (Wang et al., 2022b) | $95.33_{\pm0.06}$ | 91.07 | 89.68 | 84.08 |
| | | | IRNet (Lian et al., 2022b) | - | 91.51 | 90.76 | 86.19 |
| | | | DALI (Xu et al., 2023) | - | 92.67 | 91.83 | 89.8 |
| | | | PiCO+ w/ Mixup (Xu et al., 2023) | - | 93.56 | 92.65 | 88.21 |
| | | | DALI w/ Mixup (Xu et al., 2023) | - | 95.19 | 93.89 | 92.26 |
| | | | Ours | $\mathbf{96.28_{\pm0.13}}$ | $\mathbf{95.82_{\pm0.07}}$ | $\mathbf{95.28_{\pm0.08}}$ | $\mathbf{94.35_{\pm0.08}}$ |
| CIFAR-100 | 50,000 | 0.01 | PiCO+ (Wang et al., 2022b) | $76.29_{\pm0.42}$ | 71.42 | 70.22 | 66.14 |
| | | | IRNet (Lian et al., 2022b) | - | 71.17 | 70.10 | 68.77 |
| | | | DALI (Xu et al., 2023) | - | 72.26 | 71.98 | 71.04 |
| | | | PiCO+ w/ Mixup (Xu et al., 2023) | - | 75.04 | 74.31 | 71.79 |
| | | | DALI w/ Mixup (Xu et al., 2023) | - | 76.52 | 76.55 | 76.09 |
| | | | Ours | $\mathbf{78.08_{\pm0.26}}$ | $\mathbf{77.53_{\pm0.24}}$ | $\mathbf{76.96_{\pm0.02}}$ | $\mathbf{76.43_{\pm0.27}}$ |
| | | 0.05 | PiCO+ (Wang et al., 2022b) | $76.17_{\pm0.18}$ | 69.40 | 66.67 | 62.24 |
| | | | IRNet (Lian et al., 2022b) | - | 70.73 | 69.33 | 68.09 |
| | | | DALI (Xu et al., 2023) | - | 72.28 | 71.35 | 70.05 |
| | | | PiCO+ w/ Mixup (Xu et al., 2023) | - | 73.06 | 71.37 | 67.56 |
| | | | DALI w/ Mixup (Xu et al., 2023) | - | 76.87 | 75.23 | 74.49 |
| | | | Ours | $\mathbf{76.95_{\pm0.46}}$ | $\mathbf{77.07_{\pm0.16}}$ | $\mathbf{76.34_{\pm0.08}}$ | $\mathbf{75.13_{\pm0.63}}$ |
| | | 0.1 | PiCO+ (Wang et al., 2022b) | $75.55_{\pm0.21}$ | - | - | - |
| | | | IRNet (Lian et al., 2022b) | - | - | - | - |
| | | | DALI (Xu et al., 2023) | - | - | - | - |
| | | | PiCO+ w/ Mixup (Xu et al., 2023) | - | - | - | - |
| | | | DALI w/ Mixup (Xu et al., 2023) | - | - | - | - |
| | | | Ours | $\mathbf{76.41_{\pm1.02}}$ | $\mathbf{75.50_{\pm0.54}}$ | $\mathbf{74.67_{\pm0.30}}$ | $\mathbf{73.88_{\pm0.60}}$ |

Table 20: NLL ablation

Table 19: PLL ablation

Table 18: SSL ablation

| | CIFAR100 $l=200$ | STL10 $l=40$ |
|---|---|---|
| Ours | 22.06 | 11.09 |
| Ours w/o ent. | 22.41 | 11.23 |

| | CIFAR10 $q=0.5$ | CIFAR100 $q=0.1$ |
|---|---|---|
| PiCO | 93.58 | 69.91 |
| Ours | 95.91 | 74.00 |
| PiCO w/o s. a. | 91.78 | 66.43 |
| Ours w/o s. a. | 94.53 | 72.69 |
| Ours w/o ent. | 95.87 | 73.75 |

| | CIFAR10 $\eta=0.5$ | CIFAR100 $\eta=0.1$ |
|---|---|---|
| SOP | 94.00 | 63.30 |
| Ours | 94.31 | 66.46 |
| SOP w/o s. a. | 66.85 | 36.60 |
| Ours w/o s. a. | 93.56 | 65.89 |
| SOP w/o ent. | 93.04 | 62.85 |
| Ours w/o ent. | 94.16 | 66.12 |

Table 21: Runtime Analysis on CIFAR-100

| Setting | Algorithm | CIFAR-100 Avg. Runtime (s/iter) |
|---|---|---|
| SSL | FreeMatch | 0.2157 |
| SSL | Ours | 0.1146 |
| PLL | PiCO | 0.3249 |
| PLL | Ours | 0.2919 |
| NLL | SOP | 0.1176 |
| NLL | Ours | 0.1021 |

