# OpenReview forum: "Imprecise Label Learning: A Unified Framework for Learning with Various Imprecise Label Configurations"
_TMLR — Withdrawn by Authors_

### Review · Reviewer_BQHj · 2024-03-17

**Summary Of Contributions:**

The paper proposes a unified framework of EM-algorithm-based learning of classifiers when label information is imprecise. By imprecise information the authors mean basically any relationship between the observed imprecise label and true latent label. As extremes, the imprecise label could be equal to the latent label, or they might be completely unrelated, as if the label information were fully missing. The results demonstrate the usefulness of the proposed method in practice. In comparison with the existing methods, the proposed method performs comparably to the state-of-the-art, occasionally better and occasionally worse.

**Audience:**

Yes

**Broader Impact Concerns:**

The work has no such ethical implications that would require adding a Broader Impact Statement section.

**Claims And Evidence:**

Yes

**Requested Changes:**

In Section 3, page 5, it is confusing and mathematically incorrect to state that $f\circ g:\mathcal{X}\rightarrow\mathcal{Y}$ and later say that the output from $f\circ g$ is the predicted probability.

In Appendix C.7, clarify the comparison in the number of iterations that were run for the proposed method and other methods. Even if per-iteration speed is comparable, then it is important to complement this statement with information about the number of iterations each method was run with. I suggest moving the runtime analysis into the main part of the paper, but this is up to the authors to decide.

**Strengths And Weaknesses:**

Strengths:

While good methods for different specific settings of imprecise labels exist, the paper presents a nice unified approach.

The results demonstrate the usefulness of the proposed method in practice. In comparison with the existing methods, the proposed method performs comparably to the state-of-the-art, occasionally better and occasionally worse.

Weaknesses:

The runtime analysis could also mention the number of iterations, when comparing different methods.

In Section 3, page 5, it is confusing and mathematically incorrect to state that $f\circ g:\mathcal{X}\rightarrow\mathcal{Y}$ and later say that the output from $f\circ g$ is the predicted probability.

---

### Review · Reviewer_ots8 · 2024-03-19

**Summary Of Contributions:**

The submission aims to consider models learning from imprecise labels in a unified framework ILL, which is based on an EM for modeling imprecise label information.

**Audience:**

Yes

**Broader Impact Concerns:**

No concern.

**Claims And Evidence:**

No

**Requested Changes:**

The manuscript might need a thorough revision. First, it needs to separate previous work from its own contribution. Second, the conceptual issues are somewhat fatal and need to be addressed. Third, there should be a rigorous analysis of experiment results. In particular, it needs to show evidence that the proposed contribution leads to better performance.

**Strengths And Weaknesses:**

Strength:

1. The work has done an extensive overview of existing methods.
2. The reported performances from new implementations of the framework seem to have improvement over previous methods.

Weakness:

1. It is unclear the exact contribution of the work. There is a long history of using EM algorithm to treat learning problems with imprecise labels. In particular, true labels are treated as hidden variables, and the learning goal is to maximize the likelihood of observed imprecise labels. This approach is not new. The submission doesn't seem to have a significant contribution.

2. There are several conceptual mistakes/shortcomings in this work.

First, the formulation in (4) can directly model the conditional probability $p(I | X; \theta)$, instead of claiming that $P(X)$ does not rely on $\theta$. It is more common to consider discriminative models than generative models in classification problems.

Second, the claim "If $I$ contains information about the true labels $Y$ , such as the actual labels or the label candidates, it ($P(I | )$) can be reduced to P(I|Y )" is not accurate: it should be that the imprecise label $I$ is solely derived from $Y$, then it holds. In some cases, $I$ contains information about both $Y$ and $X$.

Third, I don't see a sense to apply it to semi-supervised cases because $Y^U$ provides no information. In equation (7), we can just set $\mathcal{A}_s = \mathcal{A}_w$ then the first term of (7) should be zero, and it does not affect the learning of the second term.

Fourth, I don't see how (8) is derived. I feel that the probability term should be multiplied by both log terms.


3. The should be more in-depth analysis how the proposed method improves the performance.

---

### Review · Reviewer_AEJH · 2024-04-21

**Summary Of Contributions:**

The paper introduces a novel framework, Imprecise Label Learning (ILL), that unifies various learning paradigms under imprecise label configurations. ILL uses Expectation-Maximization to model the imprecise label information, treats the precise labels as latent variables, and considers the distribution of all possible labeling entailed by the imprecise labels. ILL handles seamlessly three major paradigms (i.e., partial labels, semi-supervised learning, and noisy labels) and any mixture thereof. The empirical evaluations (unfortunately based solely on artificially-created datasets) shows that, compared to other approaches, ILL is competitive, and, often, both robust and effective.

**Audience:**

Yes

**Broader Impact Concerns:**

In this reviewer's opinion, this paper does NOT rise any concerns about ethical implications.

**Claims And Evidence:**

Yes

**Requested Changes:**

The paper can be significantly improved improved by (1) adding a real-world dataset,  both to motivate the approach and to anchor the evaluation, (2) adding an illustrative running example to provide an intuitive dimension to Section 4, and (3) the empirical evaluation needs tightening-up .

(1) add as motivating examples a few realistic real-world application domains, and, ideally, a real-world dataset in the evaluation. I fully agree with TMLR's policy on "emphasizing technical correctness over subjective significance, to ensure that we facilitate scientific discourse on topics that may not yet be accepted in mainstream venues but may be important in the future;" however, relying solely on mainstream datasets with synthetically-injected partial and noisy labels doesn't do justice to the paper (actually, it takes away from its merits). If nothing else, you could re-use (or replicate at a larger scale) the real-world domain from (Denoeux 2011).

(2) to increase the paper's readability and accessibility-to-wider-audiences, you should begin Section 4 with a running example that intuitively illustrates the paper's key concepts of (i) using Expectation-Maximization to model the imprecise label information, (ii) treating the precise labels as latent variables, and (iii) considering the distribution of all possible labeling entailed by the imprecise labels.

(3) the empirical evaluation needs both tightening up and adding several key details.
- in Table 1, what do you mean by the row "Fully Supervised?" the best performance for CIFAR-10 and CIFAR-100 are 99.5% and 96.08%, respectively. They are far superior to the values you list there, and they raise serious doubts about ILL's performance on CIFAR-100: why should we care about 74%-75% accuracy, when SOTA is 96.08? The same applies, to a lesser extent, to CIFAR-10, on which ILL's 96.37% doesn't even make it in the top-100 results listed in https://paperswithcode.com/sota/image-classification-on-cifar-10
- your setup from 5.1 is a great example why synthetically-created, partial-label datasets have very little value: in a real-world domain, the "partial label" scenario would typically imply some ORGANIC ambiguity in which of the remaining labels is the best/correct one. In your case, the flipped negative labels are not based on the organic ambiguity of the example, which should make it less of a challenge to identify the correct label
- why do you use accuracy in Tables 1, 3, 4, 5, but error rate in Table 2? It is confusing
- ideally, in Table 2, for each dataset, you should also have a column withe "the smallest number of labeled examples that makes the SSL algorithms competitive with a supervised one." Which algorithm does best with 200 examples is irrelevant if the error rate is orders of magnitude worse than SOTA
- add to Table 2 a "Fully Supervised" row like Table 1 has
- why did you change the "noise ratios" between tables 2 & 3?
- in Table 2, you forgot to underline ELR as the runner-up for CIFAR-100 in the "Asym" column
- in Table 5, adding a "noise = 0" column is  a fine idea; you should also add, for each value of "l" the corresponding rows for "q = 0;" last but not least, you should also add the results when training on 50% and 100% of the labeled examples

OTHERS:
- is the first paragraph of the Intro, please add a reference or example to "intrinsically difficult to ascertain precisely"
- last line on page 1 & first line on page 2: to preserve parallelism, you should provide an intuitive explanation (or definition) for "programable" and "bag-level" weak supervision, as crowd-sourcing does for "multiple (imprecise) annotations"
- Fig 1.d gives the wrong impression that, for noisy labels, ALL LABELS ARE/MUST-BE WRONG
- the last, 4-line sentence in the 2nd paragraph on page 2 is ambiguous because it lacks a ")"
- page 3, above related work: you do not define "scalable and consistent," and you don't seem to use these terms again, which makes the claim worthless
- page 4 - you must back the "... is a highly realistic scenario" by AT THE VERY LEAST a few illustrative application domains that should have been introduced in the first section; ideally, there should also be a real-world dataset in the empirical evaluation
- you must re-write the last paragraph in Section 2: it is the most relevant work to ILL, and it is treated far too shallowly. First you should add a table that, for each of the previous attempts towards unification, you provide similarities & differences with/from ILL, pros/cons of the existing approach, and their underlying assumptions. Second, in the new, re-written paragraph, you should use the narrative form to go deeper into comparing & contrasting ILL and existing approaches
- first sentence in section 4: please carefully document, intuitively explain, and add reference to support the claim of "frequently fall short in adaptability & transferability;" ideally, this should have been part of the re-writing of the last paragraph in section 3
- IMHO, the sentence "It stands poised to transform ..." is meaningless without the authors adding to the paper some real-world domains & datasets

**Strengths And Weaknesses:**

The paper tackles the important problem of learning under the limitations imposed by various types of imprecise labels. The proposed approach  appears to be novel and sound. Overall, the paper is reasonably well-written, but it could (and should) be improved to be more easily accessible to a wider audience: the underlying idea is good and worth spreading around.

The paper has three main weaknesses: (1) it lacks a real-world dataset to motivate the approach and to anchor the evaluation, (2) its  empirical evaluation needs both tightening-up and several clarifications, and (3) it lacks an illustrative running example that would add an intuitive dimension to the current Section 4. We will address all these issues in detail in the next section.

---

### Note · Authors · 2024-05-15

I have read and agree with the venue's withdrawal policy on behalf of myself and my co-authors.